# Functionally Constrained Algorithm Solves Convex Simple Bilevel Problems

**Huaqing Zhang**[*1,2]   **Lesi Chen**[*1,2]   **Jing Xu**[1]   **Jingzhao Zhang**[1,2,3]

[1]IIIS, Tsinghua University   [2]Shanghai Qizhi Institute
[3]Shanghai AI Lab

{zhanghq22, chenlc23, xujing21}@mails.tsinghua.edu.cn
jingzhaoz@mail.tsinghua.edu.cn

## Abstract

This paper studies simple bilevel problems, where a convex upper-level function is minimized over the optimal solutions of a convex lower-level problem. We first show the fundamental difficulty of simple bilevel problems, that the approximate optimal value of such problems is not obtainable by first-order zero-respecting algorithms. Then we follow recent works to pursue the weak approximate solutions. For this goal, we propose a novel method by reformulating them into functionally constrained problems. Our method achieves near-optimal rates for both smooth and nonsmooth problems. To the best of our knowledge, this is the first near-optimal algorithm that works under standard assumptions of smoothness or Lipschitz continuity for the objective functions.

## 1 Introduction

This work focuses on the following optimization problem:

$$\min_{\mathbf{x} \in \mathcal{Z}} f(\mathbf{x}) \quad \text{s.t.} \quad \mathbf{x} \in \mathcal{X}_g^* \triangleq \arg\min_{\mathbf{z} \in \mathcal{Z}} g(\mathbf{z}), \tag{1}$$

where $f, g$ are convex and continuous functions and $\mathcal{Z} \subseteq \mathbb{R}^n$ is a compact convex set. Such a problem is often referred to as "simple bilevel optimization" in the literature, as the upper-level objective function $f$ is minimized over the solution set of a lower-level problem. It captures a hierarchical structure and thus has many applications in machine learning, including lexicographic optimization [13, 15] and lifelong learning [13, 18]. Understanding the structure of simple bilevel optimization and designing efficient algorithms for it is vital and has gained massive attention in recent years [1, 5, 6, 10, 12–14, 19, 23–27].

To solve the problem, one may observe that Problem (1) is equivalent to the convex optimization problem $\min_{\mathbf{x} \in \mathcal{X}_g^*} f(\mathbf{x})$ with *implicitly* defined convex domain $\mathcal{X}_g^*$. Hence, it is natural to try to design first-order methods to find $\hat{\mathbf{x}} \in \mathbb{R}^n$ such that

$$|f(\hat{\mathbf{x}}) - f^*| \le \epsilon_f, \quad g(\hat{\mathbf{x}}) - g^* \le \epsilon_g, \tag{2}$$

where $f^*$ is the optimal value of Problem (1) and $g^*$ is the optimal value of the lower-level problem $(\min_{z \in \mathcal{Z}} g(z))$. We highlight the **asymmetry** in $f$ and $g$ here. $f^*$ is the minimal in the constrained set $\mathcal{X}_g^*$, and hence it is possible that $f(\mathbf{x}) < f^*$ for some $\mathbf{x} \in \mathcal{Z}$. On the other hand, $g^*$ is globally minimal and hence $g^* \le g(\mathbf{x})$ for any $\mathbf{x} \in \mathcal{Z}$. *Such asymmetry is natural as the role of $f, g$ are inherently asymmetrical for bilevel problems.* We call such $\hat{\mathbf{x}}$ a $(\epsilon_f, \epsilon_g)$-*absolute optimal solution*.

---

[*]Equal contributions.

When $\mathcal{X}_g^*$ is explicitly given, finding such a solution is easy as we can apply methods for constrained optimization problems such as the projected gradient method and Frank-Wolfe method [4, Section 3]. Yet, somewhat surprisingly, our first contribution in this paper (Theorem 4.1 and 4.2) shows that it is generally intractable for any zero-respecting first-order method to find absolute optimal solutions for Problem (1). Our negative result shows the fundamental difficulty of simple bilevel problems compared to classical constrained optimization problems.

As a compromise, most approaches developed for simple bilevel optimization in the literature aim to find a solution $\hat{\mathbf{x}} \in \mathbb{R}^n$ such that

$$f(\hat{\mathbf{x}}) - f^* \leq \epsilon_f, \quad g(\hat{\mathbf{x}}) - g^* \leq \epsilon_g, \tag{3}$$

which we call a $(\epsilon_f, \epsilon_g)$-*weak optimal solution*. Much progress has been achieved towards this goal [10, 18, 19, 23–25]. We note all the above algorithms fall in the class of zero-respecting algorithms (Assumption 3.4) and hence cannot obtain absolute optimal solutions, unless additional assumptions are made. (See Remark4.1 and Appendix D for further discussions.)

Our second contribution pushes this boundary by proposing near-optimal lower and upper bounds. We study two settings: **(a)** $f$ is $C_f$-Lipschitz and $g$ is $C_g$-Lipschitz, **(b)** $f$ is $L_f$-smooth and $g$ is $L_g$-smooth. We can extend the worst-case functions for single-level optimization to Problem (1) to show lower bounds of

1. $\Omega\left(\max\left\{C_f^2/\epsilon_f^2, C_g^2/\epsilon_g^2\right\}\right)$ for the setup (a);

2. $\Omega\left(\max\left\{\sqrt{L_f/\epsilon_f}, \sqrt{L_g/\epsilon_g}\right\}\right)$ for the setup (b).

Given our constructed lower bounds, we further improve known upper bounds by reducing the task of finding $(\epsilon_f, \epsilon_g)$-weak optimal solutions to minimizing the functionally constrained problem:

$$\min_{\mathbf{x} \in \mathcal{Z}} f(\mathbf{x}), \quad \text{s.t.} \quad \tilde{g}(\mathbf{x}) \triangleq g(\mathbf{x}) - \hat{g}^* \leq 0, \tag{4}$$

where $\hat{g}^*$ is an approximate solution to the lower level problem $\min_{\mathbf{x} \in \mathcal{Z}} g(\mathbf{x})$. Then we further leverage the reformulation by Nesterov [20, Section 2.3.4] which relates the optimal value of Problem (4) to the minimal root of the following auxiliary function, where a discrete minimax problem defines the function value:

$$\psi^*(t) = \min_{\mathbf{x} \in \mathcal{Z}} \left\{\psi(t, \mathbf{x}) \triangleq \max\left\{f(\mathbf{x}) - t, \tilde{g}(\mathbf{x})\right\}\right\}. \tag{5}$$

Based on this reformulation, we introduce a novel method FC-BiO (Functionally Constrained Bilevel Optimizer). FC-BiO is a double-loop algorithm. It adopts a bisection procedure on $t$ in the outer loop and applies gradient-based methods to solve the sub-problem (5). Our algorithms achieve the following upper bounds:

1. $\tilde{\mathcal{O}}\left(\max\left\{C_f^2/\epsilon_f^2, C_g^2/\epsilon_g^2\right\}\right)$ for the setup (a);

2. $\tilde{\mathcal{O}}\left(\max\left\{\sqrt{L_f/\epsilon_f}, \sqrt{L_g/\epsilon_g}\right\}\right)$ for the setup (b),

where $\tilde{\mathcal{O}}$ hides logarithmic terms. Both complexity upper bounds match the corresponding lower bounds up to logarithmic factors. In words, we summarize our contributions as follows:

- We prove the intractability for any zero-respecting first-order methods to find a $(\epsilon_f, \epsilon_g)$-absolute optimal solution of simple bilevel problems.

- We propose a novel method FC-BiO that has near-optimal rates for finding $(\epsilon_f, \epsilon_g)$-weak optimal solutions of both nonsmooth and smooth problems. To the best of our knowledge, this is the first near-optimal algorithm that works under standard assumptions of smoothness or Lipschitz continuity for the objective functions. A comparison of previous results can be found in Section 2.

## 2 Related work

In the literature, various methods [1, 3, 6, 10, 12–14, 18, 19, 23–27] have been proposed to achieve a $(\epsilon_f, \epsilon_g)$-weak optimal solution to simple bilevel problems defined as Equation (3) . Below, we review the existing methods with non-asymptotic convergence. For ease of presentation, we state the results for $\epsilon_f = \epsilon_g = \epsilon$.

**Prior results on Lipschitz problems**  Kaushik and Yousefian [14] proposed the averaging iteratively regularized gradient method (a-IRG) for convex optimization with variational inequality constraints, of which Problem (1) is a special case. a-IRG achieves the rate of $\mathcal{O}(1/\epsilon^4)$. Shen et al. [25] proposed a method for solving Problem (1) with $\mathcal{O}(1/\epsilon^3)$ complexity based on the online learning framework. When $f$ is Lipschitz continuous and $g$ is smooth, Merchav and Sabach [19] proposed a gradient-based algorithm with $\mathcal{O}(1/\epsilon^{1/(1-\alpha)})$ complexity for any $\alpha \in (0.5, 1)$. However, none of these methods can achieve the optimal rate of $\mathcal{O}(\epsilon^{-2})$.

**Prior results on smooth problems**  Samadi et al. [24] proposed the regularized accelerated proximal method (R-APM) with a complexity of $\mathcal{O}(1/\epsilon)$. Under the additional weak sharp minima condition on $g$, the complexity of R-APM improves to $\mathcal{O}\left(1/\sqrt{\epsilon}\right)$. However, this condition is often too strong and does not hold for many problems. Chen et al. [10] extended the result of [24] to the more general $\alpha$-Hölderian error bound condition. However, their method achieves the optimal rate only when $\alpha = 1$, which reduces to the weak sharp minima condition. Jiang et al. [13] developed a conditional gradient type algorithm (CG-BiO) with a complexity of $\mathcal{O}\left(1/\epsilon\right)$, which approximates $\mathcal{X}_g^*$ similar to the cutting plane approach. Later on, Cao et al. [6] proposed an accelerated algorithm with a similar cutting plane approach to achieve the rate of $\mathcal{O}(\max\{1/\sqrt{\epsilon_f}, 1/\epsilon_g\})$, which can further be improved to $\mathcal{O}(1/\sqrt{\epsilon})$ under the additional weak sharp minima condition. Recently, Wang et al. [28] reduced Problem (1) to finding the smallest $c$ such that the optimal value of the following parametric problem is $g^*$: $\min_{\mathbf{x} \in \mathbb{R}^n} g(\mathbf{x})$, s.t. $f(\mathbf{x}) \le c$. They adopted a bisection method to find such a $c$. To solve this parametric problem, Accelerated Proximal Gradient method is applied on $g$ with projection operator onto the sublevel set $\mathcal{F}_c = \{\mathbf{x} \mid f(\mathbf{x}) \le c\}$, which we call *sublevel set oracles*. This leads to an upper bound of $\tilde{\mathcal{O}}(1/\sqrt{\epsilon})$. Such an oracle is obtainable for norm-like functions such as $f(\mathbf{x}) = \frac{1}{2}\|\mathbf{x}\|^2$. However, it may be computationally intractable for more general functions, such as MSE loss or logistic loss. It is a very strong oracle that is seldom used in the literature on optimization: the single-level optimization of a function $f$ using sublevel set oracles can be completed in $\mathcal{O}(\log(1/\epsilon))$ iterations of bisection procedure. Compared with previous work [6, 10, 24, 28], our proposed methods achieve the $\tilde{\mathcal{O}}(1/\sqrt{\epsilon})$ rate *under standard assumptions, without assuming $f$ is a norm-like function or $g$ satisfies the weak sharp minima condition.*

**Comparison with Nesterov's methods for functionally constrained problems**  Based on similar reformulation, Nesterov [20] has proposed algorithms for functionally constrained problems, of which Problem (4) is a special case: one for smooth problems in Section 2.3.5, and one for Lipschitz problems in Section 3.3.4. However, Nesterov's algorithm for smooth problems relies on the strong convexity of $f$ and $g$, which does not hold in our bilevel setups. In this case, $\mathcal{X}_g^*$ would be a singleton, rendering the upper-level problem trivial. Our algorithm does not require the strong convexity assumption, and has a unified framework for both smooth and nonsmooth problems.

## 3 Preliminaries

For any $\mathbf{x} \in \mathbb{R}^n$, let $\mathbf{x}_{[j]}$ represent the $j$-th coordinate of $\mathbf{x}$ for $j = 1, \cdots, n$. We use $\operatorname{supp}(\mathbf{x}) := \{j \in [d] : \mathbf{x}_{[j]} \ne 0\}$ to denote the support of $\mathbf{x}$. The Euclidean ball centered at $\mathbf{x}$ with radius $R$ is denoted as $\mathcal{B}(\mathbf{x}, R) \triangleq \{\mathbf{y} \mid \|\mathbf{y} - \mathbf{x}\|_2 \le R\}$. For any closed convex set $\mathcal{C} \subseteq \mathbb{R}^n$, the Euclidean projection of $\mathbf{x}$ onto $\mathcal{C}$ is denoted by $\Pi_{\mathcal{C}}(\mathbf{x}) \triangleq \arg\min_{\mathbf{y} \in \mathcal{C}} \|\mathbf{y} - \mathbf{x}\|_2$. We say a function $h$ is $C$-Lipschitz in domain $\mathcal{Z}$ if $\|h(\mathbf{x}) - h(\mathbf{y})\|_2 \le C\|\mathbf{x} - \mathbf{y}\|_2$ for all $\mathbf{x}, \mathbf{y} \in \mathcal{Z}$. We say a differentiable real-valued function $h$ is $L$-smooth if it has $L$-Lipschitz continuous gradients.

We now state the assumptions required in our theoretical results.

**Assumption 3.1.** *Consider Problem (1). We assume that*

> *1. Functions $f$ and $g : \mathbb{R}^n \to \mathbb{R}$ are convex and continuous.*

*2. The feasible set $\mathcal{Z}$ is convex and compact with diameter $D = \sup_{\mathbf{x},\mathbf{y} \in \mathcal{Z}} \|\mathbf{x} - \mathbf{y}\|_2$.*

The compactness assumption ensures that the subprocesses adopted in our method have a unified upper complexity bound (see Section 5.3). We note that other works involving bisection procedures, such as Wang et al. [28], may also need to address this issue to derive an explicit dependence on the distance term (although it is not stated formally in their paper). For unconstrained problems, if we know that the initial distance $\|\mathbf{x}^* - \mathbf{x}_0\|_2$ is upper bounded by $R$, we can simply take $\mathcal{Z} = \mathcal{B}(\mathbf{x}_0, R)$.

We use Assumption 3.1 throughout this paper, but distinguish the following two different settings.

**Assumption 3.2.** *Consider Problem (1). We assume that $f$ and $g$ are $L_f$-smooth and $L_g$-smooth respectively. We call such problems $(L_f, L_g)$-smooth problems.*

**Assumption 3.3.** *Consider Problem (1). We assume that $f$ and $g$ are $C_f$-Lipschitz and $C_g$-Lipschitz in $\mathcal{Z}$ respectively. We call such problems $(C_f, C_g)$-Lipschitz problems.*

To study the complexity of solving Problem (1), we make the following assumption on the algorithms.

**Assumption 3.4** (zero-respecting algorithm class). *An iterative method $\mathcal{A}$ can access the objective functions $f$ and $g$ only through a first-order black-box oracle, which takes a test point $\hat{\mathbf{x}}$ as the input and returns $\partial f(\hat{\mathbf{x}}), \partial g(\hat{\mathbf{x}})$, where $\partial f(\hat{\mathbf{x}}), \partial g(\hat{\mathbf{x}})$ are arbitrary subgradients of the objective functions at $\hat{\mathbf{x}}$. $\mathcal{A}$ generates a sequence of test points $\{\mathbf{x}_k\}_{k=0}^K$ with*

$$\mathrm{supp}(\mathbf{x}_{k+1}) \subseteq \mathrm{supp}(\mathbf{x}_0) \cup \left( \bigcup_{0 \leq s \leq k} \mathrm{supp}\left(\partial f(\mathbf{x}_s)\right) \cup \mathrm{supp}(\partial g(\mathbf{x}_s)) \right). \tag{6}$$

This assumption generalizes the standard definition of *zero-respecting* algorithm for single-level minimization problems [7, 20]. Many existing methods that incorporate a gradient step in the update for Problem (1) clearly fall within this class of algorithms, including those proposed by [10, 18, 19, 23–25], since the gradient step ensures that $\mathbf{x}_k \in \mathbf{x}_0 + \mathrm{Span}\{\partial f(\mathbf{x}_0), \partial g(\mathbf{x}_0), \ldots, \partial f(\mathbf{x}_{k-1}), \partial g(\mathbf{x}_{k-1})\}$. In the appendix, we show that the proposed algorithm in this paper and the conditional gradient type methods [5, 13] also satisfy the condition (6) when the domain is a Euclidean ball centered at $\mathbf{x}_0$ (see Proposition C.1 and Remark C.1), which suffices to establish the negative results, including the intractability results for absolute optimal solutions and lower complexity bounds for weak optimal solutions.

The following concept of *first-order zero-chain*, introduced by Nesterov [20, Section 2.1.2], plays an essential role in proving lower bounds for zero-respecting algorithms. In our paper, we leverage the chain-like structure to show the intractability of finding absolute optimal solutions.

**Definition 3.1** (first-order zero-chain). *We call a differentiable function $h(\mathbf{x}) : \mathbb{R}^q \to \mathbb{R}$ a first-order zero-chain if for any sequence $\{\mathbf{x}_k\}_{k \geq 0}$ satisfying*

$$\mathrm{supp}(\mathbf{x}_{k+1}) \subseteq \bigcup_{0 \leq s \leq k} \mathrm{supp}\left(\nabla h(\mathbf{x}_s)\right), \; k \geq 1; \quad \mathbf{x}_0 = \mathbf{0},$$

*it holds that $\mathbf{x}_{k,[j]} = 0, k + 1 \leq j \leq q$.*

Definition 3.1 defines differentiable zero-chain functions. We can similarly define non-differentiable zero-chain, by requiring $\mathrm{supp}(\mathbf{x}_{k+1})$ to be in $\bigcup_{0 \leq s \leq k} \mathrm{supp}\left(\partial h(\mathbf{x}_s)\right)$, where $\partial h(\mathbf{x})$ is a (possibly adversarial) subgradient of $h$ at $\mathbf{x}$.

## 4 Finding absolute optimal solutions is hard

Faced with Problem (1), a natural initial response is to seek an approximate solution $\hat{\mathbf{x}}$ such that $f(\hat{\mathbf{x}})$ is as close to $f^*$ as possible, under the premise that $g(\hat{\mathbf{x}})$ is close to $g^*$. Such a goal is captured by the concept of $(\epsilon_f, \epsilon_g)$-absolute optimal solutions as defined in (2). However, it turns out that finding a $(\epsilon_f, \epsilon_g)$-absolute optimal solution is intractable for any zero-respecting first-order methods in both smooth and Lipschitz problems as shown in the following theorems.

**Theorem 4.1.** *For any first-order algorithm $\mathcal{A}$ satisfying Assumption 3.4 that runs for $T$ iterations and any initial point $\mathbf{x}_0$, there exists a $(1, 1)$-smooth instance of Problem (1) such that the optimal solution $\mathbf{x}^*$ satisfies $\|\mathbf{x}_0 - \mathbf{x}^*\|_2 \leq 1$ and $|f(\mathbf{x}_0) - f^*| \geq \frac{1}{48}$. For the iterates $\{\mathbf{x}_k\}_{k=0}^T$ generated by $\mathcal{A}$, the following holds:*

$$f(\mathbf{x}_k) = f(\mathbf{x}_0), \quad \forall 1 \leq k \leq T.$$

**Theorem 4.2.** *For any first-order algorithm $\mathcal{A}$ satisfying Assumption 3.4 that runs for $T$ iterations and any initial point $\mathbf{x}_0$, there exists a $(1,1)$-Lipschitz instance of Problem (1) and some adversarial subgradients $\{\partial f(\mathbf{x}_k), \partial g(\mathbf{x}_k)\}_{k=0}^{T-1}$ such that the optimal solution $\mathbf{x}^*$ satisfies $\|\mathbf{x}_0 - \mathbf{x}^*\|_2 \leq 1$ and $|f(\mathbf{x}_0) - f^*| \geq \frac{1}{4}$. For the iterates $\{\mathbf{x}_k\}_{k=0}^{T}$ generated by $\mathcal{A}$, the following holds*

$$f(\mathbf{x}_k) = f(\mathbf{x}_0), \quad \forall 1 \leq k \leq T.$$

The proofs of Theorem 4.1 and Theorem 4.2 rely on the concept of worst-case convex zero-chain (Proposition A.1 and A.2). We show that for any first-order zero-respecting algorithm that runs for $T$ iterations, there exists a "hard instance" such that $f(\mathbf{x}_t)$ remains unchanged from the initial value $f(\mathbf{x}_0)$ throughout the entire process. The complete proof is provided in Appendix A.

The constructions of the above hardness results are motivated by the work [9], which demonstrated that for general bilevel optimization problems of the form $\min_{\mathbf{x} \in \mathbb{R}^n, \mathbf{y} \in \mathbb{R}^m} f(\mathbf{x}, \mathbf{y})$ subject to $\mathbf{y} \in \arg\min_{\mathbf{z} \in \mathbb{R}^m} g(\mathbf{x}, \mathbf{z})$, there exists a "hard instance" in which any zero-respecting algorithm always yields $\mathbf{x}_k = \mathbf{x}_0$ for all $1 \leq k \leq T$. Although our construction has a very similar high-level idea to [9], the $f$ and $g$ we construct are different from the functions in [9] since our desired conclusion is different.

**Remark 4.1.** *Some previous works [6, 13, 24] provide guarantees for finding $(\epsilon_f, \epsilon_g)$-absolute optimal solutions. However, these works assume an additional Hölderian error bound condition [21] on $g$. Our near-optimal methods for finding weak optimal solutions, proposed in the next section, also work well under this additional assumption and achieve the best-known convergence rate for absolute suboptimality both in smooth and Lipschitz settings. See Appendix D for further discussions.*

## 5 Near-optimal methods for weak optimal solutions

Due to the intractability of obtaining $(\epsilon_f, \epsilon_g)$-absolute optimal solutions of Problem (1), most existing works focus on developing first-order methods to find $(\epsilon_f, \epsilon_g)$-weak optimal solutions as defined in (3). In this section, we establish the lower complexity bounds for finding weak optimal solutions and propose a new framework for simple bilevel problems named Functionally Constrained Bilevel Optimizer (FC-BiO) that achieves near-optimal convergence in both Lipschtitz and smooth settings.

### 5.1 Lower complexity bounds

We first establish the lower complexity bounds for finding a $(\epsilon_f, \epsilon_g)$-weak optimal solution of $(L_f, L_g)$-smooth problems and $(C_f, C_g)$-Lipschitz problems. The results follow directly from existing lower bounds for single-level optimization problems, as simple bilevel optimization is a more general framework. Although the proof is straightforward, we present the results because establishing a precise lower bound is essential for demonstrating that an algorithm is truly near-optimal.

**Theorem 5.1.** *Given $L_f, L_g, D > 0$. For any first-order algorithm $\mathcal{A}$ satisfying Assumption 3.4 and any initial point $\mathbf{x}_0$, there exists a $(L_f, L_g)$-smooth instance of Problem (1) on the domain $\mathcal{Z} = \mathcal{B}(\mathbf{x}_0, D)$ such that the optimal solution $\mathbf{x}^*$ is contained in $\mathcal{Z}$ and $\mathcal{A}$ needs at least $\Omega\left(\max\left\{\sqrt{\frac{L_f}{\epsilon_f}}, \sqrt{\frac{L_g}{\epsilon_g}}\right\} D\right)$ iterations to find a $(\epsilon_f, \epsilon_g)$-weak optimal solution.*

**Theorem 5.2.** *Given $C_f, C_g, D > 0$. For any first-order algorithm $\mathcal{A}$ satisfying Assumption 3.4 and any initial point $\mathbf{x}_0$, there exists a $(C_f, C_g)$-Lipschitz instance of Problem (1) on the domain $\mathcal{Z} = \mathcal{B}(\mathbf{x}_0, D)$ such that the optimal solution $\mathbf{x}^*$ is contained in $\mathcal{Z}$ and $\mathcal{A}$ needs at least $\Omega\left(\max\left\{\frac{C_f^2}{\epsilon_f^2}, \frac{C_g^2}{\epsilon_g^2}\right\} D^2\right)$ iterations to find a $(\epsilon_f, \epsilon_g)$-weak optimal solution.*

### 5.2 Our proposed algorithms

We now present a unified framework applicable to both smooth and Lipschitz problems. The proposed algorithms nearly match the lower complexity bounds in both settings up to logarithmic factors.

**Problem reformulation** We apply two steps of reformulation. First, we relax Problem (1) to Problem (4), where the constraint $\mathbf{x} \in \mathcal{X}_g^*$ is replaced by a relaxed functional constraint $\tilde{g}(\mathbf{x}) \triangleq g(\mathbf{x}) - \hat{g}^* \leq 0$ and $\hat{g}^*$ is an approximate solution to the lower level problem $\min_{\mathbf{x} \in \mathcal{Z}} g(\mathbf{x})$. Denoting $\hat{f}^*$ as the optimal value of Problem (4), the following lemma holds:

**Algorithm 1** Functionally Constrained Bilevel Optimizer (FC-BiO)

---

**Require:** Problem parameters $\mathbf{x}_0, D$, desired accuracy $\epsilon$, total number of iterations $T$, initial bounds $\ell, u$, and a subroutine for Problem (7) $\mathcal{M}$.

1: Set $N = \left\lceil \log_2 \frac{u-\ell}{\epsilon/2} \right\rceil$, $K = T/N$. Set $\bar{\mathbf{x}} = \mathbf{x}_0$.
2: **for** $k = 0, \cdots, N-1$ **do**
3:    Set $t = \frac{\ell+u}{2}$.
4:    Solve with the subroutine $(\hat{\mathbf{x}}_{(t)}, \hat{\psi}^*(t)) = \mathcal{M}(\bar{\mathbf{x}}, D, t, K)$.
5:    Set $\bar{\mathbf{x}} = \hat{\mathbf{x}}_{(t)}$
6:    **if** $\hat{\psi}^*(t) > \frac{\epsilon}{2}$ **then**
7:        Set $\ell = t$.
8:    **else**
9:        Set $u = t$.
10:    **end if**
11: **end for**
12: **return** $\hat{\mathbf{x}} = \hat{\mathbf{x}}_{(u)}$ as the approximate solution.

---

**Lemma 5.1.** *If $g^* \leq \hat{g}^* \leq g^* + \frac{\epsilon_g}{2}$ and $\hat{\mathbf{x}}$ is a $(\epsilon_f, \epsilon_g/2)$-weak optimal solution to Problem (4), i.e. $f(\hat{\mathbf{x}}) \leq \hat{f}^* + \epsilon_f, \tilde{g}(\hat{\mathbf{x}}) \leq \epsilon_g/2$, then $\hat{\mathbf{x}}$ is a $(\epsilon_f, \epsilon_g)$-weak optimal solution to Problem (1).*

Therefore, it suffices to pursue an approximate solution of Problem (4). Second, Problem (4) is further reduced to the problem of finding the smallest root of the following auxiliary function:

$$\psi^*(t) = \min_{\mathbf{x} \in \mathcal{Z}} \left\{ \psi(t, \mathbf{x}) \triangleq \max \left\{ f(\mathbf{x}) - t, \tilde{g}(\mathbf{x}) \right\} \right\}. \tag{7}$$

Such reformulation is introduced in Nesterov [20, Section 2.3] with the following characterization.

**Lemma 5.2** (Nesterov [20, Lemma 2.3.4]). *Let $\hat{f}^*$ be the optimal value of Problem (4), and let $\psi^*(t)$ be the auxiliary function as defined in (7). The following holds:*

1. *$\psi^*(t)$ is continuous, decreasing, and Lipschitz continuous with constant $1$.*

2. *$\hat{f}^*$ is exactly the smallest root of $\psi^*(t)$.*

**Bisection procedure**    Based on the preceding reformulation, we propose Algorithm 1 (FC-BiO), which uses a bisection procedure to estimate the smallest root of $\psi^*(\cdot)$. For now, we assume that the desired accuracy on upper-level and lower-level problems is the same, (*i.e.* $\epsilon_f = \epsilon_g = \epsilon$). Later we will show in Corollary 5.1 that we can handle the case when $\epsilon_f \neq \epsilon_g$ by simply scaling the objectives.

Algorithm 1 applies the bisection method within an initial interval $[\ell, u]$ which contains the smallest root, $\hat{f}^*$, for $N = \left\lceil \log_2 \frac{u-\ell}{\epsilon/2} \right\rceil$ iterations. Similar to [28], the initial interval can be obtained by applying single-level first-order methods. The lower bound $\ell$ is obtained by solving the global minimum of the upper-level objective $f$ over $x \in \mathcal{Z}$, while $u = f(\hat{\mathbf{x}}_g)$ serves as a valid upper bound, where $\hat{\mathbf{x}}_g$ is an approximate solution to the lower-level problem. Further details can be found in Appendix B.1. In each iteration, we set $t = \frac{\ell+u}{2}$. To approximate the function value of $\psi^*(t)$, we apply a first-order algorithm $\mathcal{M}$ to solve the discrete minimax problem (7). For the Lipschitz setting, we let $\mathcal{M}$ be the Subgradient Method (SGM, Algorithm 2) [4, Section 3.1]. For the smooth setting, we let $\mathcal{M}$ be the generalized accelerated gradient method (generalized AGM, Algorithm 3) [20, Algorithm 2.3.12]. These methods guarantee to find an approximate solution $\hat{\mathbf{x}}_{(t)}$ of Problem (7) such that

$$\psi^*(t) \leq \hat{\psi}^*(t) \triangleq \psi(t, \hat{\mathbf{x}}_{(t)}) \leq \psi^*(t) + \frac{\epsilon}{2}. \tag{8}$$

If $\hat{\psi}^*(t) > \epsilon/2$, we update $\ell = t$. Conversely, if $\hat{\psi}^*(t) \leq \epsilon/2$, we set $u = t$. For the initial point of $\mathcal{M}$, we exploit a *warm-start* strategy (see more details in Appendix B.2). After completing $N$ iterations, we return $\hat{\mathbf{x}} = \hat{\mathbf{x}}_{(u)}$ as the output. As shown in Lemma 5.3, $\hat{\mathbf{x}}$ is guaranteed to be a $(\epsilon, \epsilon)$-weak optimal solution to Problem (1).

**Algorithm 2** Solve Problem (7) with SGM $(\mathbf{x}_0, D, t, K)$

---

**Require:** Problem parameters $\mathbf{x}_0, D, t$, total number of iterations $K$.
1: Set $\eta = D/(C\sqrt{K})$, where $C = \max\{C_f, C_g\}$.
2: **for** $k = 0, \cdots, K-1$ **do**
3:     Obtain a subgradient $\mathbf{s} \in \partial_{\mathbf{x}}\psi(t, \mathbf{x})$ by Proposition 5.1.
4:     Update $\mathbf{x}_{k+1} = \Pi_{\mathcal{Z}}(\mathbf{x}_k - \eta\mathbf{s})$.
5: **end for**
6: **return** $\hat{\mathbf{x}}_{(t)} = \frac{1}{K}\sum_{k=0}^{K-1}\mathbf{x}_k$ as the approximate solution and $\hat{\psi}^*(t) = \max\{f(\hat{\mathbf{x}}_{(t)}) - t, \tilde{g}(\hat{\mathbf{x}}_{(t)})\}$ as the approximate value.

---

We remark that since we can only solve an approximate value of $\psi^*(t)$, the upper bound $u$ might fall below $\hat{f}^*$ during the bisection process. But this is acceptable since we are only in pursuit of a weak optimal solution instead of an absolute optimal solution.

**Lemma 5.3.** *If $\hat{g}^*$ satisfies $g^* \leq \hat{g}^* \leq g^* + \frac{\epsilon}{2}$ and (8) holds for every $t$ in the process of Algorithm 1, then the approximate solution $\hat{\mathbf{x}}$ returned by Algorithm 1 is a $(\epsilon, \epsilon/2)$-weak optimal solution to Problem (4), and therefore a $(\epsilon, \epsilon)$-weak optimal solution to Problem (1) by Lemma 5.1.*

According to this lemma, $\mathcal{O}(\log(1/\epsilon))$ iterations of the outer loops are sufficient to find a $(\epsilon, \epsilon)$-weak optimal solution. Next, we will discuss the process and complexity of the subroutines in detail.

### 5.3 Subroutines and total complexity

To proceed with the bisection process, we need to invoke a subroutine $\mathcal{M}$ to approximate the function value of $\psi^*(t) = \min_{\mathbf{x}\in\mathcal{Z}}\psi(t, \mathbf{x})$ in each outer iteration, where $\psi(t, \mathbf{x}) = \max\{f(\mathbf{x}) - t, \tilde{g}(\mathbf{x})\}$. This reduces to a discrete minimax optimization problem (Problem (7)) for a given $t$. Below we demonstrate the subroutines to solve this problem in Lipschitz and smooth settings.

**Lipschitz setting** When $f$ and $g$ are convex and $C_f$ and $C_g$-Lipschitz respectively, it holds that $\psi(t, \mathbf{x})$ is also convex and Lipschitz with constant $\max\{C_f, C_g\}$. In this case, setting $\mathcal{M}$ to be the Subgradient Method (SGM) [4, Section 3.1] applied on $\psi(t, \cdot)$ (Algorithm 2) directly achieves the optimal convergence rate. To implement the SGM method, the subgradient of $\psi(t, \mathbf{x})$ needs to be computed as given in the following proposition:

**Proposition 5.1** (Nesterov [20, Lemma 3.1.13]). *Consider $\psi(t, \mathbf{x}) = \max\{f(\mathbf{x}) - t, \tilde{g}(\mathbf{x})\}$ where $f$ and $g$ are convex functions. For given $t$. We have*

$$\partial_{\mathbf{x}}\psi(t, \mathbf{x}) = \begin{cases} \partial f(\mathbf{x}), & f(\mathbf{x}) - t > \tilde{g}(\mathbf{x}); \\ \partial\tilde{g}(\mathbf{x}), & f(\mathbf{x}) - t < \tilde{g}(\mathbf{x}); \\ \mathrm{Conv}\{\partial f(\mathbf{x}), \partial\tilde{g}(\mathbf{x})\}, & f(\mathbf{x}) - t = \tilde{g}(\mathbf{x}). \end{cases}$$

Algorithm 2 has the following convergence guarantee:

**Lemma 5.4** (Bubeck et al. [4, Theorem 3.2]). *Suppose Assumption 3.1 and 3.3 hold. When $K \geq \frac{4D^2C^2}{\epsilon^2}$, the approximate value $\hat{\psi}^*(t)$ produced by Algorithm 2 satisfies $\psi^*(t) \leq \hat{\psi}^*(t) \leq \psi^*(t) + \frac{\epsilon}{2}$, where $C = \max\{C_f, C_g\}$.*

We refer to Algorithm 1 with SGM subroutine (Algorithm 2) as FC-BiO$^{\mathtt{Lip}}$. Combining with Lemma 5.3, we obtain the following total iteration complexity of FC-BiO$^{\mathtt{Lip}}$:

**Theorem 5.3** (Lipschitz setting). *Suppose Assumption 3.1 and 3.3 hold and $\epsilon_f = \epsilon_g = \epsilon$. When*

$$T \geq \left\lceil\log_2\frac{u - \ell}{\epsilon/2}\right\rceil\frac{4\max\{C_f^2, C_g^2\}}{\epsilon^2}D^2,$$

*the approximate solution $\hat{\mathbf{x}}$ produced by FC-BiO$^{Lip}$ is a $(\epsilon, \epsilon)$-weak optimal solution to Problem (1).*

**Smooth setting** The optimal first-order method for optimizing smooth objective functions is the celebrated Accelerated Gradient Method (AGM) [20, Section 2.2] proposed by Nesterov. In contrast to the Lipschitz setting, AGM cannot be applied to $\psi(t, \mathbf{x}) = \max\{f(\mathbf{x}) - t, \tilde{g}(\mathbf{x})\}$ directly when

**Algorithm 3** Solve Problem (7) with Generalized AGM $(\mathbf{x}_0, D, t, K)$ [20, Algorithm 2.3.12]

---

**Require:** Problem parameters $\mathbf{x}_0, D, t$, total number of iterations $K$.
1: Set $\mathbf{y}_0 = \mathbf{x}_0$, $\alpha_0 = \frac{1}{2}$.
2: **for** $k = 0, \cdots K - 1$ **do**
3:     Compute $\mathbf{x}_{k+1}$ as the solution to (9) by Proposition 5.2.
4:     Compute $\alpha_{k+1}$ from the equation $\alpha_{k+1}^2 = (1 - \alpha_{k+1})\alpha_k^2$.
5:     Set $\beta_k = \frac{\alpha_k(1-\alpha_k)}{\alpha_k^2 + \alpha_{k+1}}$, $\mathbf{y}_{k+1} = \mathbf{x}_{k+1} + \beta_k(\mathbf{x}_{k+1} - \mathbf{x}_k)$.
6: **end for**
7: **return** $\hat{\mathbf{x}}_{(t)} = \mathbf{x}_K$ as the approximate solution and $\hat{\psi}^*(t) = \max\{f(\mathbf{x}_K) - t, \tilde{g}(\mathbf{x}_K)\}$ as the approximate value.

---

$f$ and $g$ are convex and smooth, as the smoothness condition no longer holds for $\psi(t, \cdot)$. However, Nesterov [20, Section 2.3] showed that by simply replacing the gradient step in standard AGM with the following *gradient mapping* (Nesterov [20, Definition 2.3.2])

$$\mathbf{x}_{k+1} = \arg\min_{\mathbf{x} \in \mathcal{Z}} \left\{ \bar{\psi}(t, \mathbf{x}; \mathbf{y}_k) \triangleq \max \left\{ f(\mathbf{y}_k) + \langle \nabla f(\mathbf{y}_k), \mathbf{x} - \mathbf{y}_k \rangle + \frac{L}{2}\|\mathbf{x} - \mathbf{y}_k\|_2^2 - t, \right.\right.$$
$$\left.\left. \tilde{g}(\mathbf{y}_k) + \langle \nabla \tilde{g}(\mathbf{y}_k), \mathbf{x} - \mathbf{y}_k \rangle + \frac{L}{2}\|\mathbf{x} - \mathbf{y}_k\|_2^2 \right\} \right\}, \tag{9}$$

the optimal rate of $\mathcal{O}(\sqrt{L/\epsilon})$ can be achieved for Problem (7). Here $\{\mathbf{x}_k\}$, $\{\mathbf{y}_k\}$ are the test point sequences and $L = \max\{L_f, L_g\}$. Solving $\mathbf{x}_{k+1}$ for general discrete minimax problems (where the maximum is taken over potentially more than two objective functions, as studied in Nesterov [20, Section 2.2]), reduces to a quadratic programming (QP) problem and may not be efficiently solvable. However, we demonstrate that in our problem setup, $\mathbf{x}_{k+1}$ can be expressed in the form of a projection onto the feasible set $\mathcal{Z}$, or onto the intersection of $\mathcal{Z}$ and a hyperplane. A similar subproblem also arises in Cao et al. [6, Remark 3.2]. When the structure of $\mathcal{H}$ is simple, such as when it is a Euclidean ball, the subproblem may admit a closed-form solution. Otherwise, Dykstra's projection algorithm can be applied to solve it [11].

**Proposition 5.2.** *Define the descent step candidates*

$$\mathbf{x}_1 = \Pi_{\mathcal{Z}}\left(\mathbf{y}_k - \frac{1}{L}\nabla f(\mathbf{y}_k)\right), \quad \mathbf{x}_2 = \Pi_{\mathcal{Z}}\left(\mathbf{y}_k - \frac{1}{L}\nabla \tilde{g}(\mathbf{y}_k)\right),$$
$$\mathbf{x}_3 = \Pi_{\mathcal{Z} \cap \mathcal{H}}\left(\mathbf{y}_k - \frac{1}{L}\nabla f(\mathbf{y}_k)\right), \tag{10}$$

*where $\mathcal{H} \subset \mathbb{R}^n$ is a hyperplane defined by*

$$\mathcal{H} = \{\mathbf{x} \mid f(\mathbf{y}_k) - \tilde{g}(\mathbf{y}_k) + \langle \nabla f(\mathbf{y}_k) - \nabla \tilde{g}(\mathbf{y}_k), \mathbf{x} - \mathbf{y}_k \rangle - t = 0\}.$$

*Then the solution to (9) is $\mathbf{x}_{k+1} = \arg\min_{\{\mathbf{x}_i | i \in \{1,2,3\}\}} \bar{\psi}(t, \mathbf{x}_i; \mathbf{y}_k)$.*

We present the generalized AGM subroutine (Algorithm 3) and its convergence rate below.

**Lemma 5.5** (Nesterov [20, Theorem 2.3.5]). *Suppose Assumption 3.1 and 3.2 hold. When $K \geq D\sqrt{\frac{12L}{\epsilon}}$, the approximate value $\hat{\psi}^*(t)$ produced by Algorithm 3 satisfies $\psi^*(t) \leq \hat{\psi}^*(t) \leq \psi^*(t) + \frac{\epsilon}{2}$, where $L = \max\{L_f, L_g\}$.*

We refer to Algorithm 1 with generalized AGM subroutine (Algorithm 3) as FC-BiO$^{\text{sm}}$, whose total iteration complexity is given by the following theorem:

**Theorem 5.4** (Smooth setting). *Suppose Assumption 3.1 and 3.2 hold and $\epsilon_f = \epsilon_g = \epsilon$. When*

$$T \geq \left\lceil \log_2 \frac{u - \ell}{\epsilon/2} \right\rceil \sqrt{\frac{12 \max\{L_f, L_g\}}{\epsilon}} D,$$

*the approximate solution $\hat{\mathbf{x}}$ produced by FC-BiO$^{\text{sm}}$ is a $(\epsilon, \epsilon)$-weak optimal solution to Problem (1).*

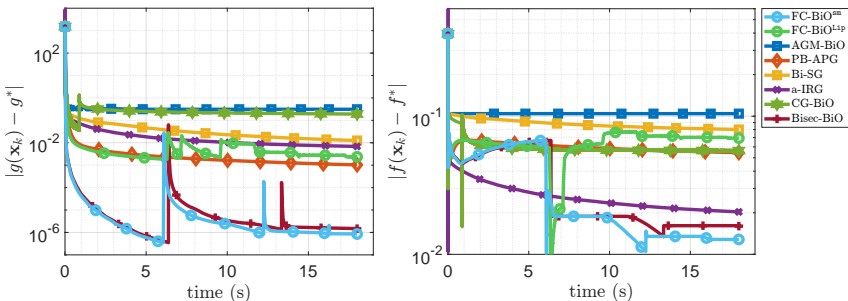

Figure 1: The performance of Algorithm 1 compared with other methods in Problem (11).

For more general cases when the desired accuracy for the upper-level and lower-level problems is different (*i.e.* $\epsilon_f \neq \epsilon_g$), we can simply scale the objective functions before applying FC-BiO$^{\tt Lip}$ or FC-BiO$^{\tt sm}$, resulting in the following guarantee:

**Corollary 5.1.** *Suppose Assumption 3.1 holds. By scaling $\tilde{g}^\circ = \frac{\epsilon_f}{\epsilon_g}\tilde{g}$ and applying FC-BiO$^{Lip}$ or FC-BiO$^{sm}$ on functions $f$ and $\tilde{g}^\circ$, a $(\epsilon_f, \epsilon_g)$-weak optimal solution to Problem (1) is obtained within the complexity of $\tilde{\mathcal{O}}\left(\max\left\{\frac{C_f^2}{\epsilon_f^2}, \frac{C_g^2}{\epsilon_g^2}\right\} D^2\right)$ under Assumption 3.3 and $\tilde{\mathcal{O}}\left(\max\left\{\sqrt{\frac{L_f}{\epsilon_f}}, \sqrt{\frac{L_g}{\epsilon_g}}\right\} D\right)$ under Assumption 3.2.*

Our proposed algorithms are near-optimal in both Lipschitz and smooth settings as the convergence results in Corollary 5.1 match the lower bounds established in Theorem 5.1 and Theorem 5.2.

## 6 Numerical experiments

In this section, we evaluate our proposed methods on two different bilevel problems with smooth objectives. We compare the performance of FC-BiO$^{\tt sm}$ with existing methods, including a-IRG [14], Bi-SG [19], CG-BiO[13], AGM-BiO [6], PB-APG[10], and Bisec-BiO[28]. The following problems are also Lipschitz on a compact set $\mathcal{Z}$, so we implement FC-BiO$^{\tt Lip}$ as well. The initialization time of FC-BiO$^{\tt sm}$, FC-BiO$^{\tt Lip}$, CG-BiO, and Bisec-BiO is taken into account and is plotted in the figures. Our implementations of CG-BiO and a-IRG are based on the codes from [13], which is available online[1].

### 6.1 Minimum norm solution

As in [28], we consider the following simple bilevel problem:

$$f(\mathbf{x}) = \frac{1}{2}\|\mathbf{x}\|_2^2, \quad g(\mathbf{x}) = \frac{1}{2}\|A\mathbf{x} - b\|_2^2. \tag{11}$$

We set the feasible set $\mathcal{Z} = \mathcal{B}(\mathbf{0}, 2)$. We use the Wikipedia Math Essential dataset [22], which contains 1068 instances with 730 attributes. We uniformly sample 400 instances and denote the feature matrix and outcome vector by $A$ and $\mathbf{b}$ respectively. We choose the same random initial point $\mathbf{x}_0$ for all methods. We set $\epsilon_f = \epsilon_g = 10^{-6}$. For this problem, we can explicitly solve $\mathbf{x}^*$ and $f^*$ to measure the convergence. Figure 1 shows the superior performance of our method compared to existing methods in both upper-level and lower-level. The only exception is Bisec-BiO [28], which shows a comparable performance to our method. This also aligns well with the theory as these two methods have the same convergence rates. We remark that the output of our method satisfies that $f(\hat{\mathbf{x}}) < f^*$. Thus although $|f(\hat{\mathbf{x}}) - f^*| > \epsilon_f$, indeed a $(\epsilon_f, \epsilon_g)$-weak optimal solution is solved by FC-BiO$^{\tt sm}$. See more experiment details in Appendix E.1.

### 6.2 Over-parameterized logistic regression

We examine simple bilevel problems where the lower-level and upper-level objectives correspond to the training loss and validation loss respectively. Here we address the logistic regression problem

---

[1]`https://github.com/Raymond30/CG-BiO`

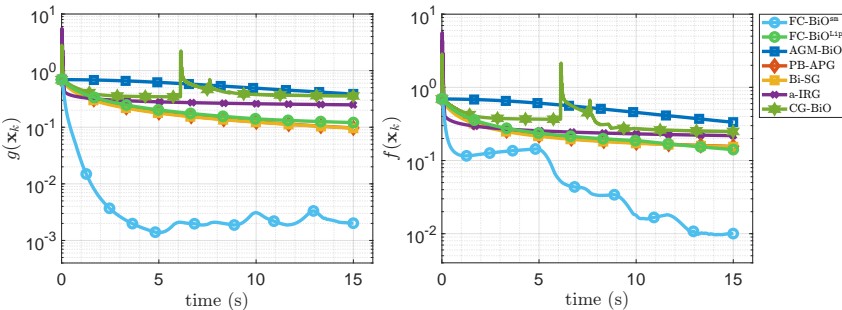

Figure 2: The performance of Algorithm 1 compared with other methods in Problem (12)

using the "rcv1.binary" dataset from "LIBSVM" [8, 16], which contains $20,242$ instances with $47,236$ features. We uniformly sample $m = 5000$ instances as the training dataset $(A^{tr}, \mathbf{b}^{tr})$, and $m$ instances as the validation dataset $(A^{val}, \mathbf{b}^{val})$. We consider the bilevel problem with:

$$
\begin{aligned}
f(\mathbf{x}) &= \frac{1}{m} \sum_{i=1}^{m} \log(1 + \exp(-(A_i^{val})^\top \mathbf{x} \mathbf{b}_i^{val})), \\
g(\mathbf{x}) &= \frac{1}{m} \sum_{i=1}^{m} \log(1 + \exp(-(A_i^{tr})^\top \mathbf{x} \mathbf{b}_i^{tr})).
\end{aligned}
\tag{12}
$$

We set the feasible set $\mathcal{Z} = \mathcal{B}(\mathbf{0}, 300)$. We set the initial point $\mathbf{x}_0 = \mathbf{0}$ for all methods. We set $\epsilon_f = \epsilon_g = 10^{-3}$. Since projecting to the sublevel set of $f(\mathbf{x})$ is not practical, Bisec-BiO [28] does not apply to this problem, thus we only consider other methods. To the best of our knowledge, no existing solver could obtain the exact value of $f^*$ and $g^*$ of Problem (12). Thus we only plot function value $f(\mathbf{x}_k)$ and $g(\mathbf{x}_k)$, instead of suboptimality. As shown in Figure 2, our method converges faster than other algorithms in both upper-level and lower-level. More details are provided in Appendix E.2.

## 7  Conclusion and future work

This paper provides a comprehensive study of convex simple bilevel problems. We show that finding a $(\epsilon_f, \epsilon_g)$-absolute optimal solution for such problems is intractable for any zero-respecting first-order algorithm, thus justifying the notion of weak optimal solution considered by existing works. We then propose a novel method FC-BiO for finding a $(\epsilon_f, \epsilon_g)$-weak optimal solution. Our proposed method achieves the near-optmal rates of $\tilde{\mathcal{O}}\left(\max\{L_f^2/\epsilon_f^2, L_g^2/\epsilon_g^2\}\right)$ and $\tilde{\mathcal{O}}\left(\max\{\sqrt{L_f/\epsilon_f}, \sqrt{L_g/\epsilon_g}\}\right)$ for Lipschitz and smooth problems respectively. To the best of our knowledge, this is the first near-optimal algorithm that works under standard assumptions of smoothness or Lipschitz continuity for the objective functions.

We discuss some limitations unaddressed in this work. First, our method introduces an additional logarithmic factor compared to the lower bounds. We hope future works can further close this gap between upper and lower bounds. Second, our methods cannot be directly applied to stochastic problems [5]. Establishing lower complexity bounds and developing optimal methods for stochastic simple bilevel problems remain an open question for future research.

## Acknowledgement

We would like to express our sincere gratitude to the anonymous reviewers and the area chair for their invaluable feedback and insightful suggestions. In particular, we thank the area chair for suggesting simplifying the proofs of Theorem 5.1 and Theorem 5.2.

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

# A  Proofs for Section 4

## A.1  Constructions of first-order zero-chains

We first give the constructions of the zero-chains that are used in our negative results. These zero-chains are the worst-case functions that Nesterov [20] used to prove the lower complexity bounds for first-order methods in single-level optimization.

**Proposition A.1** (Paraphrased from Section 2.1.2 Nesterov [20])**.** *Consider the family of functions* $h_{q,L,R}(\mathbf{x}) : \mathbb{R}^q \to \mathbb{R}$:

$$h_{q,L,R}(\mathbf{x}) = \frac{L}{4}\left(\frac{1}{2}\left(\mathbf{x}_{[1]}^2 + \sum_{i=1}^{q-1}(\mathbf{x}_{[i]} - \mathbf{x}_{[i+1]})^2 + \mathbf{x}_{[q]}^2\right) - R\mathbf{x}_{[1]}\right) \tag{13}$$

*The following properties hold for any* $h_{q,L,R}(\mathbf{x})$ *with* $q \in \mathbb{N}^+$ *and* $L, R > 0$:

1. *It is a convex and L-smooth function.*

2. *It has the following unique minimizer:* $\mathbf{x}_{[j]}^* = R\left(1 - \frac{j}{q+1}\right)$ *with norm* $\|\mathbf{x}^*\|_2 \leq R\sqrt{q}$.

3. *It is a differentiable first-order zero-chain as defined in Definition 3.1.*

**Remark A.1.** *In Section 2.1.2 of Nesterov [20], it is shown that* $h_{q,L,1}$ *is a convex and L-smooth zero-chain. Here we define* $h_{q,L,R}(\mathbf{x}) := R^2 h_{q,L,1}(\frac{\mathbf{x}}{R})$. *Then* $\nabla^2 h_{q,L,R}(\mathbf{x}) = \nabla^2 h_{q,L,1}(\frac{\mathbf{x}}{R})$, *and* $h_{q,L,R}(\mathbf{x})$ *is also a convex and L-smooth zero-chain. Similarly, in Proposition A.2, we define* $r_{q,C,R}(\mathbf{x}) := Rr_{q,C,1}(\frac{\mathbf{x}}{R})$. *Nesterov [20] showed* $r_{q,C,1}$ *is a C-Lipschitz zero-chain, which implies that* $r_{q,C,R}$ *is also a C-Lipschitz zero-chain, since* $\nabla r_{q,C,R}(\mathbf{x}) = \nabla r_{q,C,R}(\frac{\mathbf{x}}{R})$.

**Proposition A.2** (Paraphrased from Section 3.2.1 Nesterov [20])**.** *Consider the family of functions* $r_{q,L,R}(\mathbf{x}) : \mathbb{R}^q \to \mathbb{R}$:

$$r_{q,C,R}(\mathbf{x}) = \frac{C\sqrt{q}}{1+\sqrt{q}}\max_{1\leq j\leq q}\mathbf{x}_{[j]} + \frac{C}{2R(1+\sqrt{q})}\|\mathbf{x}\|_2^2. \tag{14}$$

*The following properties hold for any* $r_{q,L,R}(\mathbf{x})$ *with* $q \in \mathbb{N}^+$ *and* $C, R > 0$:

1. *It is a convex function and is C-Lipschitz in the Euclidean ball* $\mathcal{B}(\mathbf{0}, R)$.

2. *It has a unique minimizer* $\mathbf{x}^* = -\frac{R}{\sqrt{q}}\mathbf{1}$.

3. *It is a non-differentiable first-order zero-chain.*

**Remark A.2.** *The non-differentiable first-order zero-chain presented in Proposition A.2 is slightly different from the one given in Nesterov [20].* $r_{q,C,R}(\cdot)$ *is Lipschitz in* $\mathcal{B}(\mathbf{0}, R)$, *while the zero-chain presented in Nesterov [20] is Lipschitz in* $\mathcal{B}(\mathbf{x}^*, R)$, *where* $\mathbf{x}^*$ *is the minimizer of the zero-chain.*

## A.2  Proofs of Theorem 4.1 and Theorem 4.2

*Proof of Theorem 4.1.* Consider any first-order algorithm $\mathcal{A}$ satisfying Assumption 3.4 that runs for $T$ iterations. Without loss of generality, we assume the initial point of $\mathcal{A}$ is $\mathbf{x}_0 = \mathbf{0}$. Otherwise, we can translate the following construction to $f(\mathbf{x} - \mathbf{x}_0), g(\mathbf{x} - \mathbf{x}_0)$. Let $q = 2T$ and

$$f(\mathbf{x}) = \frac{1}{2}\sum_{j=T+1}^{q}\mathbf{x}_{[j]}^2, \quad g(\mathbf{x}) = h_{q,1,1/\sqrt{q}}(\mathbf{x}),$$

where $h_{q,1,1/\sqrt{q}}(\cdot)$ follows Proposition A.1. It is clear from the construction that both $f(\cdot), g(\cdot)$ are convex and 1-smooth. Furthermore, the optimal solution to Problem (1) defined by such $f, g$ is the unique minimizer of $g(\mathbf{x})$, given by $\mathbf{x}_{[j]}^* = \frac{1}{\sqrt{q}}(1 - \frac{j}{q+1})$. We prove by induction on $k$ that the test points $\{\mathbf{x}_k\}_{k=0}^T$ generated by $\mathcal{A}$ satisfies $\mathbf{x}_{k,[j]} = 0$ for $0 \leq k \leq T, T+1 \leq j \leq 2T$: Suppose for some $k \leq T, \mathbf{x}_{i,[j]} = 0$ holds for $0 \leq i \leq k-1, T+1 \leq j \leq 2T$, we have

$$\nabla f(\mathbf{x}_i) = 0, \quad 0 \leq i \leq k-1.$$

The zero-respecting assumption of $\mathcal{A}$ (Assumption 3.4) leads to

$$\mathbf{x}_k \in \bigcup_{0 \leq i \leq k-1} \mathrm{supp}(\nabla f(\mathbf{x}_i)) \cup \mathrm{supp}(\nabla g(\mathbf{x}_i))$$

$$= \bigcup_{0 \leq i \leq k-1} \mathrm{supp}(\nabla g(\mathbf{x}_i)).$$

Since $g(\mathbf{x})$ is a first-order zero-chain, we conclude that

$$\mathbf{x}_{k,[j]} = 0, \quad k+1 \leq j \leq 2T.$$

Consequently, $f(\mathbf{x}_k)$ remains zero for all $0 \leq k \leq K$. However,

$$f^* = \frac{1}{2} \sum_{j=T+1}^{2T} \mathbf{x}_{[j]}^{*2}$$

$$= \frac{1}{4T} \sum_{j=T+1}^{2T} \left( 1 - \frac{j}{2T+1} \right)^2$$

$$= \frac{1}{24} \cdot \frac{T+1}{2T+1}$$

$$\geq \frac{1}{48}.$$

Thus

$$|f(\mathbf{x}_k) - f^*| \geq \frac{1}{48}, \quad 1 \leq k \leq T.$$

$\square$

*Proof of Theorem 4.2.* Consider any first-order algorithm $\mathcal{A}$ satisfying Assumption 3.4 that runs for $T$ iterations. We similarly assume that the initial point of $\mathcal{A}$ is $\mathbf{x}_0 = 0$. Let $q = 2T$ and

$$f(\mathbf{x}) = \frac{1}{2} \sum_{j=T+1}^{2T} \mathbf{x}_{[j]}^2, \ g(\mathbf{x}) = r_{2T,1,1}(\mathbf{x}),$$

where $r_{2T,1,1}(\cdot)$ follows the construction in Proposition A.2. Both $g(\cdot)$ and $f(\cdot)$ are convex and 1-Lipschitz in $\mathcal{B}(\mathbf{0}, 1)$, and the unique minimizer $\mathbf{x}^* = -\frac{1}{\sqrt{q}}\mathbf{1}$ of $g$ is the optimal solution to Problem (1) defined by such $f$ and $g$, with norm $\|\mathbf{x}^*\|_2 = 1$. Similar to the proof of Theorem 4.1, we prove by induction on $k$ that there exist some adversarial subgradients $\{\partial g(\mathbf{x}_0), \cdots, \partial g(\mathbf{x}_{k-1})\}$ such that the test points $\{\mathbf{x}_k\}_{k=0}^T$ generated by $\mathcal{A}$ satisfies $\mathbf{x}_{k,[j]} = 0$ for $0 \leq k \leq T, T+1 \leq j \leq 2T$: Suppose for some $k \leq T$, $\mathbf{x}_{i,[j]} = 0$ holds for $0 \leq i \leq k-1, T+1 \leq j \leq 2T$, we have

$$\nabla f(\mathbf{x}_i) = 0, \quad 0 \leq i \leq k-1.$$

The zero-respecting assumption of $\mathcal{A}$ (Assumption 3.4) leads to

$$\mathbf{x}_k \in \bigcup_{0 \leq i \leq k-1} \mathrm{supp}(\nabla f(\mathbf{x}_i)) \cup \mathrm{supp}(\partial g(\mathbf{x}_i))$$

$$= \bigcup_{0 \leq i \leq k-1} \mathrm{supp}(\partial g(\mathbf{x}_i)),$$

where $\{\partial g(\mathbf{x}_0), \cdots, \partial g(\mathbf{x}_{k-1})\}$ are the subgradients returned by a black-box first-order oracle. Since $g(\mathbf{x})$ is a first-order zero-chain, we conclude that there exists some adversarial subgradients $\{\partial g(\mathbf{x}_0), \cdots, \partial g(\mathbf{x}_{k-1})\}$ such that

$$\mathbf{x}_{k,[j]} = 0, \quad k+1 \leq j \leq 2T.$$

Consequently, $f(\mathbf{x}_k)$ remains zero for all $0 \leq k \leq K$. However, $f^* = f(\mathbf{x}^*) = \frac{1}{4}$. Thus

$$|f(\mathbf{x}_k) - f^*| \geq \frac{1}{4}, \quad 1 \leq k \leq T.$$

$\square$

# B    Algorithm details

## B.1    Determining $\hat{g}^*$ and the initial interval

To apply the two-step reformulation, we need to solve an approximate value of the lower-level problem $\hat{g}^*$ such that $g^* \leq \hat{g}^* \leq g^* + \frac{\epsilon}{2}$. Furthermore, to conduct the bisection method of FC-BiO, it is necessary to first determine an initial interval $[\ell, u]$ that contains the minimal root of $\psi^*(t)$, which is exactly $\hat{f}^*$ (Lemma 5.2). The following procedure is inspired by Wang et al. [28]: First, we apply optimal first-order methods for single-level optimization problems – subgradient method (SGM) [4, Section 3.1] for Lipschitz functions, and accelerated gradient method (AGM) [20, Section 2.2] for smooth functions – to the lower-level objective $g$ to obtain an approximate minimum point $\hat{\mathbf{x}}_g \in \mathbb{R}^n$ such that $g^* \leq g(\hat{\mathbf{x}}_g) \leq g^* + \frac{\epsilon}{2}$. We set $\hat{g}^* = g(\hat{\mathbf{x}}_g)$ and define the relaxed constraint function $\tilde{g}(\mathbf{x})$ as in (4). We set $u = f(\hat{\mathbf{x}}_g)$, then

$$\psi^*(u) \leq \psi(u, \hat{\mathbf{x}}_g) = \max\{f(\hat{\mathbf{x}}_g) - u, \tilde{g}(\hat{\mathbf{x}}_g)\} = 0.$$

Thus $u$ is indeed an upper-bound of the minimal root of $\psi^*(\cdot)$ given that $\psi^*(\cdot)$ is decreasing.

To derive a lower bound of the minimal root $\hat{f}^*$, we can also apply the optimal single-level minimization methods to the upper-level objective $f$ to find an approximate global minimum point $\hat{\mathbf{x}}_f \in \mathcal{Z}$ such that $p^* \leq \hat{p}^* \triangleq f(\hat{\mathbf{x}}_f) \leq p^* + \frac{\epsilon}{2}$, where $p^*$ is the minimum value of $f$ over $\mathcal{Z}$ (i.e., $p^* \triangleq \min_{\mathbf{x} \in \mathcal{Z}} f(\mathbf{x})$). Then $\ell = \hat{p}^* - \frac{\epsilon}{2}$ is a valid lower bound of $\hat{f}^*$ since

$$\ell = \hat{p}^* - \frac{\epsilon}{2} \leq p^* \leq \hat{f}^*.$$

Nevertheless, any lower bound for $\hat{f}^*$ is acceptable, such as 0 when the upper-level objective $f$ is non-negative.

The complexity of applying the optimal single-level minimization methods to $f$ and $g$ separately is $\mathcal{O}\left(\frac{C_f^2}{\epsilon^2}D^2 + \frac{C_g^2}{\epsilon^2}D^2\right)$ (SGM for Lipschitz problems) or $\mathcal{O}\left(\sqrt{\frac{L_f}{\epsilon}}D + \sqrt{\frac{L_g}{\epsilon}}D\right)$ (AGM for smooth problems), which does not increase the total complexity established in Theorem 5.3 and 5.4.

## B.2    Warm-start strategy

For the initial point of the subroutine $\mathcal{M}$ (SGM in FC-BiO$^{\texttt{Lip}}$ or generalized AGM in FC-BiO$^{\texttt{Sm}}$ ), we exploit the *warm-start* strategy, which uses the last point in the previous round (denoted by $\bar{\mathbf{x}}$ in Algorithm 1) as the initial point of the current round. Intuitively, the subproblem (*i.e.* minimizing $\max\{f(\mathbf{x}) - t, g(\mathbf{x})\}$) does not change significantly in each round since the parameter $t$ does not change too much as $\ell$ and $u$ are getting closer. Thus the approximate solution of the last round is supposed to be close to the optimal solution of the current round. Nevertheless, any initial point in $\mathcal{Z}$ does not change the complexity upper bound. Such a strategy is widely used in two-level optimization methods [2, 9, 17].

Similarly, we can use the approximate minimum point $\hat{\mathbf{x}}_g$ of $g$ (as described in Appendix B.1) instead of $\mathbf{x}_0$ as the initial point of the first round of the subroutine.

# C    Proofs for Section 5

## C.1    Proofs of Theorem 5.1 and Theorem 5.2

Recall the lower complexity bounds for single-level convex optimization problems:

**Lemma C.1** (Nesterov [20, Theorem 2.1.7]). *Given $L > 0, D > 0$. For any first-order algorithm $\mathcal{A}$ satisfying Assumption 3.4 and any initial point $\mathbf{x}_0$, there exists a convex and $L$-smooth function $f$ such that the minimizer $\mathbf{x}^*$ satisfies $\|\mathbf{x}^* - \mathbf{x}_0\|_2 \leq D$ and*

$$f(\mathbf{x}_T) - f(\mathbf{x}^*) \geq \frac{3LD^2}{32(T+1)^2},$$

**Lemma C.2** (Nesterov [20, Theorem 3.2.1]). *Given $C > 0, D > 0$. For any first-order algorithm $\mathcal{A}$ satisfying Assumption 3.4 and any initial point $\mathbf{x}_0$, there exists a convex function $f$ that is $C$-Lipschitz in $\mathcal{B}(\mathbf{x}_0, D)$, such that the minimizer $\mathbf{x}^*$ satisfies $\|\mathbf{x}^* - \mathbf{x}_0\|_2 \leq D$ and*

$$f(\mathbf{x}_T) - f(\mathbf{x}^*) \geq \frac{CD}{2(1 + \sqrt{T})},$$

The "hard functions" $f(\cdot)$ in the previous lemmas are exactly the zero-chain $h_{2T+1,L,D}(\cdot)$ and $r_{T,C,D}(\cdot)$ as constructed in Proposition A.1 and Proposition A.2, respectively.

*Proof of Theorem 5.1.* Consider any first-order algorithm $\mathcal{A}$ satisfying Assumption 3.4 that runs for $T$ iterations. Without loss of generality, we assume the initial point of $\mathcal{A}$ is $\mathbf{x}_0 = \mathbf{0}$. Otherwise, we can translate the following construction to $f(\mathbf{x} - \mathbf{x}_0), g(\mathbf{x} - \mathbf{x}_0)$.

If $\frac{L_f}{\epsilon_f} \geq \frac{L_g}{\epsilon_g}$, we set the upper-level and lower-level objective $f, g : \mathbb{R}^{2T+1} \to \mathbb{R}$ be:

$$f(\mathbf{x}) = h_{2T+1,L_f,D}(\mathbf{x}), \quad g(\mathbf{x}) = 0,$$

where $h_{2T+1,L_f,D}(\cdot)$ is defined in Proposition A.1. Then the zero-respecting assumption on $\mathcal{A}$ implies that for any $k \geq 1$, we have $\mathrm{supp}(\mathbf{x}_{k+1}) \subseteq \bigcup_{0 \leq s \leq k-1} \mathrm{supp}\left(\nabla f(\mathbf{x}_s)\right)$. From Lemma C.1, it holds that

$$\epsilon_f = f(\mathbf{x}_T) - f^* \geq \frac{3L_f D^2}{32(T+1)^2},$$

which implies that any zero-respecting first-order method needs at least

$$T = \Omega\left(\sqrt{\frac{L_f}{\epsilon_f}} D\right) = \Omega\left(\max\left\{\sqrt{\frac{L_f}{\epsilon_f}}, \sqrt{\frac{L_g}{\epsilon_g}}\right\} D\right)$$

iterations to solve a $(\epsilon_f, \epsilon_g)$-weak optimal solution.

If $\frac{L_f}{\epsilon_f} \leq \frac{L_g}{\epsilon_g}$, we set the upper-level and lower-level objective $f, g : \mathbb{R}^{2T+1} \to \mathbb{R}$ be:

$$f(\mathbf{x}) = 0, \quad g(\mathbf{x}) = h_{2T+1,L_g,D}(\mathbf{x}).$$

By similar arguments, any zero-respecting first-order method needs at least

$$T = \Omega\left(\sqrt{\frac{L_g}{\epsilon_g}} D\right) = \Omega\left(\max\left\{\sqrt{\frac{L_f}{\epsilon_f}}, \sqrt{\frac{L_g}{\epsilon_g}}\right\} D\right)$$

iterations to solve a $(\epsilon_f, \epsilon_g)$-weak optimal solution. $\qquad\square$

*Proof of Theorem 5.2.* Consider any first-order algorithm $\mathcal{A}$ satisfying Assumption 3.4 that runs for $T$ iterations. Without loss of generality, we assume the initial point of $\mathcal{A}$ is $\mathbf{x}_0 = \mathbf{0}$. Otherwise, we can translate the following construction to $f(\mathbf{x} - \mathbf{x}_0), g(\mathbf{x} - \mathbf{x}_0)$.

If $\frac{C_f}{\epsilon_f} \geq \frac{C_g}{\epsilon_g}$, we set the upper-level and lower-level objective $f, g : \mathbb{R}^T \to \mathbb{R}$ be:

$$f(\mathbf{x}) = r_{T,C_f,D}(\mathbf{x}), \quad g(\mathbf{x}) = 0,$$

where $r_{T,C_f,D}(\cdot)$ is defined in Proposition A.2. Then the zero-respecting assumption on $\mathcal{A}$ implies that for any $k \geq 1$, we have $\mathrm{supp}(\mathbf{x}_k) \subseteq \bigcup_{0 \leq s \leq k-1} \mathrm{supp}\left(\partial f(\mathbf{x}_s)\right)$. From Lemma C.2, it holds that

$$\epsilon_f = f(\mathbf{x}_T) - f^* \geq \frac{C_f D}{2(1 + \sqrt{T})},$$

which implies that any zero-respecting first-order method needs at least

$$T = \Omega\left(\frac{C_f^2}{\epsilon_f^2} D^2\right) = \Omega\left(\max\left\{\frac{C_f^2}{\epsilon_f^2}, \frac{C_g^2}{\epsilon_g^2}\right\} D^2\right)$$

iterations to solve a $(\epsilon_f, \epsilon_g)$-weak optimal solution.

If $\frac{C_f}{\epsilon_f} \leq \frac{C_g}{\epsilon_g}$, we set the upper-level and lower-level objective $f, g : \mathbb{R}^T \to \mathbb{R}$ be:

$$f(\mathbf{x}) = 0, \quad g(\mathbf{x}) = r_{T, C_g, D}(\mathbf{x}).$$

By similar arguments, any zero-respecting first-order method needs at least

$$T = \Omega\left(\frac{C_g^2}{\epsilon_g^2} D^2\right) = \Omega\left(\max\left\{\frac{C_f^2}{\epsilon_f^2}, \frac{C_g^2}{\epsilon_g^2}\right\} D^2\right)$$

iterations to solve a $(\epsilon_f, \epsilon_g)$-weak optimal solution.

$\square$

## C.2   Proofs in Section 5.2 and 5.3

*Proof of Lemma 5.1.*  Any feasible solution to Problem (1) is also a feasible solution to Problem (4), thus $\hat{f}^* \leq f^*$. For any $\mathbf{x}$ that satisfies

$$f(\mathbf{x}) \leq \hat{f}^* + \epsilon_f, \tilde{g}(\mathbf{x}) \leq \frac{\epsilon_g}{2},$$

we have

$$f(\mathbf{x}) \leq \hat{f}^* + \epsilon_f \leq f^* + \epsilon_f,$$
$$g(\mathbf{x}) = \tilde{g}(\mathbf{x}) + \hat{g}^* \leq \frac{\epsilon_g}{2} + g^* + \frac{\epsilon_g}{2} = g^* + \epsilon_g.$$

Thus $\mathbf{x}$ is indeed a $(\epsilon_f, \epsilon_g)$-weak optimal solution to Problem (1).

$\square$

*Proof of Lemma 5.3.*  We first show that $\ell$ is always a lower bound of $\hat{f}^*$. The initial lower bound satisfies $\ell \leq \hat{f}^*$. Furthermore, if (8) holds during the bisection process, we always have that $\psi^*(\ell) \geq \hat{\psi}^*(\ell) - \frac{\epsilon}{2} > 0$. Given that $\psi^*(\cdot)$ is decreasing, and considering that $\hat{f}^*$ is the minimal root of $\psi^*$ (Lemma 5.2), it follows that $\ell < \hat{f}^*$.

As for the upper bound $u$, the initial upper bound satisfies

$$\hat{\psi}^*(u) = \psi(u, \hat{\mathbf{x}}_g) = \max\{f(\hat{\mathbf{x}}_g) - u, \tilde{g}(\hat{\mathbf{x}}_g)\} = \max\{0, 0\} \leq \frac{\epsilon}{2}.$$

And during the bisection process, we also have that $\hat{\psi}^*(u) = \max\{f(\hat{\mathbf{x}}_{(u)}) - u, \tilde{g}(\hat{\mathbf{x}}_{(u)})\} \leq \frac{\epsilon}{2}$.

After $N = \left\lceil \log_2 \frac{u - \ell}{\epsilon/2} \right\rceil$ bisection iterations, it holds that

$$u - \ell \leq \frac{\epsilon}{2}.$$

Consider the output of Algorithm 1 $\hat{\mathbf{x}} = \hat{\mathbf{x}}_{(u)}$. Combining previous inequalities, we have

$$f(\hat{\mathbf{x}}) \leq u + \frac{\epsilon}{2} \leq \ell + \epsilon \leq \hat{f}^* + \epsilon,$$
$$\tilde{g}(\hat{\mathbf{x}}) \leq \frac{\epsilon}{2}.$$

$\square$

Some of the lemmas in this section are adapted from existing results [4, 20], but not exactly the same. We also provide a proof for these lemmas (Lemma 5.2, 5.4, 5.5).

*Proof for Lemma 5.2.*  It's clear the $\psi^*(t)$ is continuous and decreasing. For any $t \in \mathbb{R}$ and $\Delta > 0$, we have

$$\psi^*(t + \Delta) = \min_{\mathbf{x} \in \mathcal{Z}}\{\max\{f(\mathbf{x}) - t, \tilde{g}(\mathbf{x}) + \Delta\}\} - \Delta$$

$$\geq \min_{\mathbf{x} \in \mathcal{Z}} \{\max\{f(\mathbf{x}) - t, \tilde{g}(\mathbf{x})\}\} - \Delta$$
$$= \psi^*(t) - \Delta.$$

Thus $\psi^*(t)$ is 1-Lipschitz.

Let $\hat{\mathbf{x}}^*$ be the optimal solution to Problem (4), then $\hat{f}^* = f(\hat{\mathbf{x}}^*)$. For any $t \geq \hat{f}^*$, it holds that

$$\psi^*(t) \leq \psi^*(\hat{f}^*) \leq \psi(\hat{f}^*, \hat{\mathbf{x}}^*) = \max\{f(\hat{\mathbf{x}}^*) - \hat{f}^*, \tilde{g}(\hat{\mathbf{x}}^*)\} \leq 0.$$

Suppose that $t < \hat{f}^*$ and $\psi^*(t) \leq 0$, then there exists a $\hat{\mathbf{x}} \in \mathcal{Z}$ such that $\tilde{g}(\hat{\mathbf{x}}) \leq 0, f(\hat{\mathbf{x}}) - t \leq 0$. Then $\hat{\mathbf{x}}$ is a feasible solution to Problem (4), with $f(\hat{\mathbf{x}}) \leq t < \hat{f}^*$, contradiction to the fact that $\hat{f}^*$ is the optimal value to Problem (4). Thus for any $t < \hat{f}^*$, it holds that $\psi^*(t) > 0$.

Thus $\hat{f}^*$ is the smallest root of $\psi^*(t)$. $\qquad\square$

*Proof for Lemma 5.4.* Theorem 3.2 in Bubeck et al. [4] states that applying subgradient method to any $C$-Lipschitz convex function $h$ on a compact set $Q$ with diameter $D$ gives

$$h\left(\frac{1}{K}\sum_{i=0}^{K-1} \mathbf{x}_i\right) - h^* \leq \frac{DC}{\sqrt{K}}.$$

Here $h(\mathbf{x}) = \psi(t, \mathbf{x}) = \max(f(\mathbf{x}) - t, \tilde{g}(\mathbf{x})), C = \max(C_f, C_g)$. Then when $K \geq \frac{4D^2C^2}{\epsilon^2}$, it holds that

$$\hat{\psi}^*(t) = \psi\left(t, \frac{1}{K}\sum_{i=0}^{K-1}\mathbf{x}_i\right) \leq \psi^*(t) + \frac{\epsilon}{2}.$$

$\qquad\square$

*Proof for Lemma 5.5.* Theorem 2.3.5 in Nesterov [20] originally states that applying genearlized-gradient method to any $\mu$-strongly convex and $L$-smooth convex function $h$ gives

$$h(\mathbf{x}_K) - h^* \leq \frac{4L}{(\gamma_0 - \mu)(K+1)^2}(f(\mathbf{x}_0) - f^* + \frac{\gamma_0}{2}\|\mathbf{x}_0 - \mathbf{x}^*\|_2^2),$$

where $\gamma_0 = \frac{\alpha_0(\alpha_0 L - \mu)}{1 - \alpha_0}$. Here $h(\mathbf{x}) = \psi(t, \mathbf{x}) = \max(f(\mathbf{x}) - t, \tilde{g}(\mathbf{x})), L = \max(L_f, L_g), \mu = 0$, $\alpha_0 = \frac{1}{2}, \gamma_0 = \frac{L}{2}$.

Then

$$\psi(t, \mathbf{x}_K) - \psi^* \leq \frac{4L}{\frac{1}{2}L(K+1)^2}\left(\frac{L}{2}\|\mathbf{x}_0 - \mathbf{x}^*\|_2^2 + \frac{L}{4}\|\mathbf{x}_0 - \mathbf{x}^*\|_2^2\right) \leq \frac{6LD^2}{K^2}.$$

Then when $K \geq \sqrt{\frac{12L}{\epsilon}}D$, it holds that

$$\hat{\psi}^*(t) = \psi(t, \mathbf{x}_K) \leq \psi^*(t) + \frac{\epsilon}{2}.$$

$\qquad\square$

To prove Proposition 5.2, we first present a lemma:

**Lemma C.3.** *For any strictly convex and continuous functions $f, g : \mathbb{R}^n \to \mathbb{R}$, we define $\varphi : \mathbb{R}^n \to \mathbb{R}$ by $\varphi(\mathbf{x}) = \max\{f(\mathbf{x}), g(\mathbf{x})\}$. Let $\mathbf{x}^*$ be the unique minimizer of $\varphi(\cdot)$ on a convex and compact set $\mathcal{Z} \subset \mathbb{R}^n$, i.e. $\mathbf{x}^* = \arg\min_{\mathbf{x} \in \mathcal{Z}} \varphi(\mathbf{x})$, and let $\mathbf{x}_f^*, \mathbf{x}_g^*$ be the unique minimizer of $f(\cdot)$ and $g(\cdot)$ on $\mathcal{Z}$ respectively. If $\mathbf{x}^* \neq \mathbf{x}_f^*$ and $\mathbf{x}^* \neq \mathbf{x}_g^*$, then $f(\mathbf{x}^*) = g(\mathbf{x}^*)$.*

*Proof.* Below, we show that $f(\mathbf{x}^*) \neq g(\mathbf{x}^*)$ leads to contradiction. Without loss of generality, assume $f(\mathbf{x}^*) > g(\mathbf{x}^*)$. Due to the continuity of $f$ and $g$, there exists some $\delta > 0$ such that for any $0 < \theta < \delta$,

$$f(\mathbf{x}^* + \theta(\mathbf{x}_f^* - \mathbf{x}^*)) > g(\mathbf{x}^* + \theta(\mathbf{x}_f^* - \mathbf{x}^*)).$$

Furthermore, for any $\theta \in (0,1)$, we have that $\mathbf{x}^* + \theta(\mathbf{x}_f^* - \mathbf{x}^*) \in \mathcal{Z}$ and

$$f(\mathbf{x}^* + \theta(\mathbf{x}_f^* - \mathbf{x}^*)) < \theta f(\mathbf{x}_f^*) + (1-\theta)f(\mathbf{x}^*) < f(\mathbf{x}^*),$$

since $f$ is strictly convex.

Then for any $\theta \in (0, \min(1, \delta))$,

$$\varphi(\mathbf{x}^* + \theta(\mathbf{x}_f^* - \mathbf{x}^*)) = f(\mathbf{x}^* + \theta(\mathbf{x}_f^* - \mathbf{x}^*)) < f(\mathbf{x}^*) = \varphi(\mathbf{x}^*).$$

That contradicts the fact that $\mathbf{x}^*$ is the minimizer of $\varphi(\cdot)$. Thus $f(\mathbf{x}^*) = g(\mathbf{x}^*)$. $\qquad\square$

The previous lemma demonstrates that the minimizer of $\max_{\mathbf{x}\in\mathcal{Z}}\{f(\mathbf{x}), g(\mathbf{x})\}$ falls into one of three categories: the minimizer of $f$, the minimizer of $g$, or the case where the function values of $f$ and $g$ are the same. $\mathbf{x}_1, \mathbf{x}_2, \mathbf{x}_3$ in Proposition 5.2 corresponds to the three cases respectively.

*Proof for Proposition 5.2.* Define

$$\bar{f}(t, \mathbf{x}; \mathbf{y}_k) \triangleq f(\mathbf{y}_k) + \langle \nabla f(\mathbf{y}_k), \mathbf{x} - \mathbf{y}_k \rangle + \frac{L}{2}\|\mathbf{x} - \mathbf{y}_k\|_2^2 - t,$$

$$\bar{g}(\mathbf{x}; \mathbf{y}_k) \triangleq \tilde{g}(\mathbf{y}_k) + \langle \nabla g(\mathbf{y}_k), \mathbf{x} - \mathbf{y}_k \rangle + \frac{L}{2}\|\mathbf{x} - \mathbf{y}_k\|_2^2,$$

which can be written as

$$\bar{f}(t, \mathbf{x}; \mathbf{y}_k) = f(\mathbf{y}_k) - t - \frac{1}{2L}\|\nabla f(\mathbf{y}_k)\|^2 + \frac{L}{2}\left\|\mathbf{x} - \mathbf{y}_k + \frac{1}{L}\nabla f(\mathbf{y}_k)\right\|_2^2,$$

$$\bar{g}(\mathbf{x}; \mathbf{y}_k) = \tilde{g}(\mathbf{y}_k) - \frac{1}{2L}\|\nabla g(\mathbf{y}_k)\|^2 + \frac{L}{2}\left\|\mathbf{x} - \mathbf{y}_k + \frac{1}{L}\nabla g(\mathbf{y}_k)\right\|_2^2.$$

Thus

$$\mathbf{x}_1 = \arg\min_{\mathbf{x}\in\mathcal{Z}}\left\|\mathbf{x} - \mathbf{y}_k + \frac{1}{L}\nabla f(\mathbf{y}_k)\right\|_2 = \arg\min_{\mathbf{x}\in\mathcal{Z}}\bar{f}(t, \mathbf{x}; \mathbf{y}_k),$$

$$\mathbf{x}_2 = \arg\min_{\mathbf{x}\in\mathcal{Z}}\left\|\mathbf{x} - \mathbf{y}_k + \frac{1}{L}\nabla g(\mathbf{y}_k)\right\|_2 = \arg\min_{\mathbf{x}\in\mathcal{Z}}\bar{g}(t, \mathbf{x}; \mathbf{y}_k).$$

Assume the minimizer $\mathbf{x}_{k+1}$ of $\bar{\psi}(t, \mathbf{x}; \mathbf{y}_k)$ satisfies $\mathbf{x}_{k+1} \neq \mathbf{x}_1$ and $\mathbf{x}_{k+1} \neq \mathbf{x}_2$, then Lemma C.3 implies that

$$\begin{aligned}
\mathbf{x}_{k+1} \in &\{\mathbf{x} \mid \bar{f}(t, \mathbf{x}; \mathbf{y}_k) = \bar{g}(\mathbf{x}; \mathbf{y}_k)\} \\
= &\{\mathbf{x} \mid f(\mathbf{y}_k) - \tilde{g}(\mathbf{y}_k) + \langle \nabla f(\mathbf{y}_k) - \nabla\tilde{g}(\mathbf{y}_k), \mathbf{x} - \mathbf{y}_k \rangle - t = 0\} \\
= &\mathcal{H}.
\end{aligned}$$

Given that $\bar{f}(t, \mathbf{x}; \mathbf{y}_k) = \bar{g}(\mathbf{x}; \mathbf{y}_k)$ in the subset $\mathcal{Z} \cap \mathcal{H}$, we have

$$\begin{aligned}
\mathbf{x}_{k+1} &= \arg\min_{\mathbf{x}\in\mathcal{Z}\cap\mathcal{H}}\bar{\psi}(t, \mathbf{x}; \mathbf{y}_k) \\
&= \arg\min_{\mathbf{x}\in\mathcal{Z}\cap\mathcal{H}}\bar{f}(t, \mathbf{x}; \mathbf{y}_k) \\
&= \arg\min_{\mathbf{x}\in\mathcal{Z}\cap\mathcal{H}}\left\|\mathbf{x} - \mathbf{y}_k + \frac{1}{L}\nabla f(\mathbf{y}_k)\right\|_2 \\
&= \mathbf{x}_3.
\end{aligned}$$

Thus we conclude that $\mathbf{x}_{k+1}$ can only be one of $\mathbf{x}_1, \mathbf{x}_2, \mathbf{x}_3$. $\qquad\square$

*Proof for Theorem 5.3.* Combining Lemma 5.3 and Lemma 5.4, we directly get the result. $\qquad\square$

*Proof for Theorem 5.4.* Combining Lemma 5.3 and Lemma 5.5, we directly get the result. $\qquad\square$

*Proof for Corollary 5.1.* A $(\epsilon_f, \epsilon_g)$-weak optimal solution to Problem (4) is equivalent to a $(\epsilon_f, \epsilon_f)$-weak optimal solution to

$$\min_{\mathbf{x} \in \mathcal{Z}} f(\mathbf{x}), \quad \text{s.t.} \quad \tilde{g}^\circ(\mathbf{x}) = \frac{\epsilon_f}{\epsilon_g} \tilde{g}(\mathbf{x}) \le 0. \tag{15}$$

When $f$ and $g$ are $C_f, C_g$-Lipschitz respectively, $\tilde{g}^\circ$ is $\frac{\epsilon_f}{\epsilon_g} C_g$-Lipschitz. According to Theorem 5.3, FC-BiO$^{\texttt{Lip}}$ finds a $(\epsilon_f, \epsilon_f)$-weak optimal solution of Problem (15) in

$$T = \tilde{\mathcal{O}} \left( \frac{\max\{C_f^2, \frac{\epsilon_f^2}{\epsilon_g^2} C_g^2\}}{\epsilon_f^2} D^2 \right) = \tilde{\mathcal{O}} \left( \max\left\{ \frac{C_f^2}{\epsilon_f^2}, \frac{C_g^2}{\epsilon_g^2} \right\} D^2 \right)$$

iterations. Similarly, when $f$ and $g$ are $L_f, L_g$-smooth respectively, $\tilde{g}^\circ$ is $\frac{\epsilon_f}{\epsilon_g} L_g$-smooth. According to Theorem 5.3, FC-BiO$^{\texttt{sm}}$ finds a $(\epsilon_f, \epsilon_f)$-weak optimal solution of Problem (15) in

$$T = \tilde{\mathcal{O}} \left( \sqrt{\frac{\max\{L_f, \frac{\epsilon_f}{\epsilon_g} L_g\}}{\epsilon_f}} D \right) = \tilde{\mathcal{O}} \left( \max\left\{ \sqrt{\frac{L_f}{\epsilon_f}}, \sqrt{\frac{L_g}{\epsilon_g}} \right\} D \right)$$

iterations. $\qquad\square$

Finally, we prove the claim that our proposed algorithms are zero-respecting algorithms when the domain is a Euclidean ball centered at $\mathbf{x}_0$.

**Proposition C.1.** *Both FC-BiO$^{Lip}$ and FC-BiO$^{sm}$ on the domain $\mathcal{B}(\mathbf{x}_0, D)$ satisfy Assumption 3.4.*

*Proof.* Without loss of generality, we assume $\mathbf{x}_0 = \mathbf{0}$. Note that the warm-start strategy preserves the zero-respecting property in Assumption 3.4. Therefore, it suffices to prove that the subroutines (Algorithm 2 and Algorithm 3) satisfy Assumption 3.4.

For Algorithm 2, the subgradient $\mathbf{s} \in \partial_{\mathbf{x}} \psi(t, \mathbf{x})$ lies in $\text{Span}\{\partial f(\mathbf{x}), \partial g(\mathbf{x})\}$ according to Proposition 5.1. And the projection onto $\mathcal{B}(\mathbf{0}, D)$ does not disrupt the zero-respecting property. Thus, Algorithm 2 satisfies Assumption 3.4.

For Algorithm 3, we only need to prove that the gradient mapping $\mathbf{x}_{k+1}$ satisfies

$$\mathbf{x}_{k+1} \in \text{Span}\{\nabla f(\mathbf{x}_0), \nabla g(\mathbf{x}_0), \cdots, \nabla f(\mathbf{x}_k), \nabla g(\mathbf{x}_k)\}.$$

We denote $S_k = \text{Span}\{\nabla f(\mathbf{x}_0), \nabla g(\mathbf{x}_0), \cdots, \nabla f(\mathbf{x}_k), \nabla g(\mathbf{x}_k)\}$. According to Proposition 5.2, it suffices to prove that the three descent step candidates $\mathbf{x}_1, \mathbf{x}_2$ and $\mathbf{x}_3$ are in $S_k$. It is clear that

$$\mathbf{y}_k - \frac{1}{L} \nabla f(\mathbf{y}_k), \mathbf{y}_k - \frac{1}{L} \nabla \tilde{g}(\mathbf{y}_k) \in S_k.$$

Then after projection onto $\mathcal{B}(\mathbf{0}, D)$, $\mathbf{x}_1, \mathbf{x}_2$ are still in $S_k$.

The case for $\mathbf{x}_3$ is slightly more complicated. Let $\mathbf{z} = \mathbf{y}_k - \frac{1}{L} \nabla f(\mathbf{y}_k)$, $\mathbf{w} = \nabla f(\mathbf{y}_k) - \nabla \tilde{g}(\mathbf{y}_k)$, and $b = f(\mathbf{y}_k) - \tilde{g}(\mathbf{y}_k) - \langle \mathbf{w}, \mathbf{y}_k \rangle - t$. Then $\mathcal{H} = \{\mathbf{x} \mid \mathbf{w}^T \mathbf{x} + b = 0\}$. The point $\mathbf{x}_3$ is the solution to the following convex optimization problem:

$$\min_{\mathbf{x} \in \mathbb{R}^n} \quad \|\mathbf{x} - \mathbf{z}\|^2$$
$$\text{s.t.} \quad \|\mathbf{x}\|^2 \le D^2$$
$$\mathbf{w}^T \mathbf{x} + b = 0.$$

The Lagrangian for this problem is:

$$\mathcal{L}(\mathbf{x}, \lambda, \mu) = \|\mathbf{x} - \mathbf{z}\|^2 + \lambda(\|\mathbf{x}\|^2 - D^2) + \mu(\mathbf{w}^\top \mathbf{x} - b)$$

where $\lambda \ge 0$ and $\mu$ are dual variables. The KKT conditions give:

$$\nabla_{\mathbf{x}} \mathcal{L}(\mathbf{x}, \lambda, \mu)|_{\mathbf{x}=\mathbf{x}_3} = 2(\mathbf{x}_3 - \mathbf{z}) + 2\lambda \mathbf{x}_3 + \mu \mathbf{w} = 0.$$

Thus

$$\mathbf{x}_3 = \frac{2\mathbf{z} - \mu \mathbf{w}}{2(1 + \lambda)},$$

implying that $\mathbf{x}_3$ is the linear combination of $\mathbf{z}$ and $\mathbf{w}$. Since $\mathbf{z}, \mathbf{w} \in S_k$, it follows that $\mathbf{x}_3 \in S_k$. $\quad\square$

**Remark C.1.** *By similar analysis, we can show that the conditional gradient type methods for simple bilevel problems [5, 13] on domain $\mathcal{B}(\mathbf{x}_0, D)$ also fall into the zero-respecting function class. At each iteration $k$, their methods relies on the following linear program as an oracle.*

$$\mathbf{s}_k = \arg\min_{\mathbf{s}\in\mathcal{X}_k}\langle\nabla f(\mathbf{x}_k),\mathbf{s}\rangle,$$

*where $\mathcal{X}_k = \{\mathbf{s}\in\mathcal{Z} : \langle\nabla g(\mathbf{x}_k),\mathbf{s}-\mathbf{x}_k\rangle \leq g(\mathbf{x}_0)-g(\mathbf{x}_k)\}$. According to our proof of Proposition C.1, it suffices to prove that $\mathbf{s}_k$ is a linear combination of $\nabla f(\mathbf{x}_K)$ and $\nabla g(\mathbf{x}_k)$ then the remaining proofs are the same. Similarly, this can be seen by the KKT condition when $\mathcal{Z} = \mathcal{B}(\mathbf{0}, D)$, which is*

$$\nabla f(\mathbf{x}_k) + 2\lambda\mathbf{x} + \mu\nabla g(\mathbf{x}_k) = 0,$$

*where $\lambda \geq 0$ and $\mu \geq 0$ are dual variables.*

# D   Finding absolute optimal solutions under additional assumptions

In Section 4, we showed that, in general, it is intractable for any zero-respecting first-order methods to find an absolute optimal solution of Problem (1). However, it is possible to establish a lower bound for $f(\mathbf{x}_k) - f^*$ under additional assumptions. Hölderian error bound [21] is a well-studied regularity condition in the optimization literature and is utilized by previous works to establish the convergence rate of finding absolute optimal solutions [6, 13, 24]. Prior to our work, the best-known result is established by Cao et al. [6], whose method achieves a $(\epsilon_f, \epsilon_g)$-absolute optimal solution for smooth problems in $\tilde{\mathcal{O}}\left(\max\left\{1/\epsilon_f^{\frac{2\alpha-1}{2}}, 1/\epsilon_g^{\frac{2\alpha-1}{2\alpha}}\right\}\right)$ iterations. Below we will show that our proposed methods also work well with such an additional assumption and achieve superior convergence rates with regard to absolute suboptimality.

**Assumption D.1.** *The lower-level objective $g$ satisfies the Hölderian error bound condition for some $\alpha \geq 1$ and $\beta > 0$, i.e.*

$$\frac{\beta}{\alpha}\mathrm{dist}(\mathbf{x}, \mathcal{X}_g^*)^\alpha \leq g(\mathbf{x}) - g^*, \quad \forall\mathbf{x}\in\mathbb{R}^n,$$

*where $\mathrm{dist}(\mathbf{x}, \mathcal{X}_g^*) \triangleq \inf_{\mathbf{y}\in\mathcal{X}_g^*}\|\mathbf{x}-\mathbf{y}\|$ for arbitrary norm $\|\cdot\|$.*

Intuitively, when the lower-level suboptimality $g(\hat{\mathbf{x}}) - g^*$ is small, $\hat{\mathbf{x}}$ should be close to $\mathcal{X}_g^*$ if Hölderian error bound condition holds for $g$. Then we can lower bound $f(\hat{\mathbf{x}}) - f^*$ by the convexity of $f$. Jiang et al. [13] formalizes the idea in the following proposition:

**Proposition D.1** (Jiang et al. [13, Proposition 4.1]). *Assume that $f$ is convex and $g$ satisfies Assumption D.1. Define $M = \max_{\mathbf{x}\in\mathcal{X}_g^*}\{\|\nabla f(\mathbf{x})\|_*\}$ where $\|\cdot\|_*$ is the dual norm of $\|\cdot\|$. Then it holds that*

$$f(\hat{\mathbf{x}}) - f^* \geq -M\left(\frac{\alpha(g(\hat{\mathbf{x}}) - g^*)}{\beta}\right)^{\frac{1}{\alpha}}$$

*for any $\hat{\mathbf{x}}\in\mathbb{R}^n$.*

This proposition shows that when $g(\hat{\mathbf{x}})-g^* \leq \left(\frac{1}{M}\epsilon_f\right)^\alpha\frac{\beta}{\alpha}$, it holds that $f(\hat{\mathbf{x}})-f^* \geq -\epsilon_f$. Combining with Corollary 5.1, we obtain:

**Corollary D.1.** *Suppose Assumption 3.2 or Assumption 3.3 hold and $g$ satisfies Assumption D.1. FC-BiO$^{Lip}$ and FC-BiO$^{sm}$ find a $(\epsilon_f, \epsilon_g)$-absolute optimal solution within the complexity of $\tilde{\mathcal{O}}\left(\max\left\{\frac{C_f^2}{\epsilon_f^2}, \frac{C_g^2}{\epsilon_g^2}, \frac{C_g^2 M^{2\alpha}\alpha^2}{\beta^2\epsilon_f^{2\alpha}}\right\}D^2\right)$ and $\tilde{\mathcal{O}}\left(\max\left\{\sqrt{\frac{L_f}{\epsilon_f}}, \sqrt{\frac{L_g}{\epsilon_g}}, \sqrt{\frac{L_g M^\alpha\alpha}{\beta\epsilon_f^\alpha}}\right\}D\right)$ for Lipschitz and smooth problems, respectively.*

To our knowledge, this is the first result that establishes a convergence rate concerning absolute suboptimality for Lipschitz problems. In a smooth setting, our result of $\tilde{O}\left(\max\left\{1/\epsilon_f^{\frac{\alpha}{2}}, 1/\epsilon_g^{\frac{1}{2}}\right\}\right)$ is also superior to the convergence rate reported by Cao et al. [6] in both upper-level and lower-level.

# E   Experiment details

In this section, we provide more details of numerical experiments in Section 6. All experiments are implemented using MATLAB R2022b on a PC running Windows 11 with a 12th Gen Intel(R) Core(TM) i7-12700H CPU (2.30 GHz) and 16GB RAM.

### E.1 Problem (11)

This problem is a $(L_f, L_g)$-smooth problem with $L_f = 1, L_g = \lambda_{\max}(A^T A)$.

**Experiment setting**  The original Wikipedia Math Essential dataset [22] contains 1068 instances with 730 attributes. Following the setting of Jiang et al. [13] and Cao et al. [6], we randomly choose one of the columns as the outcome vector and let the rest be the new feature matrix. We uniformly sample 400 instances to make the lower-level regression problem over-parameterized. In this case, the upper-level problem is actually equivalent to

$$\min_{\mathbf{x} \in \mathcal{Z}} \frac{1}{2}\|\mathbf{x}\|_2^2 \quad \text{s.t.} \quad A\mathbf{x} = \mathbf{b}.$$

The optimal solution $\mathbf{x}^*$ for this problem can be explicitly solved via the Lagrange multiplier method:

$$\begin{pmatrix} A & O \\ I & A^T \end{pmatrix} \begin{pmatrix} \mathbf{x}^* \\ \nu \end{pmatrix} = \begin{pmatrix} \mathbf{b} \\ \mathbf{0} \end{pmatrix},$$

where $\nu$ is the Lagrange multiplier. Then we use $f^* = \frac{1}{2}\|\mathbf{x}^*\|_2^2, g^* = 0$ as the benchmark.

**Implementation details**  To be fair, all algorithms start from the same random point $\mathbf{x}_0$ of unit length as the initial point. For our Algorithm 1, we take a slightly different implementation, that instead of setting the maximum number of iterations of the inner subroutine to be $T' = T/N$, we preset $T' = 8000$. If current $\mathbf{x}_k$ already satisfies $\psi(t, \mathbf{x}_k) \leq \frac{\epsilon}{2}$, then terminate the inner subroutine directly. We adopt the warm-start strategy as described in Appendix B.2. We set $L = 0$ since $f(\mathbf{x})$ is nonnegative. For FC-BiO$^{\text{Lip}}$, we set $\eta = 3 \times 10^{-4}$. For AGM-BiO, we set $\gamma = 1$ as in [6, Theorem 4.1]. For PB-APG, we set $\gamma = 10^5$. For Bi-SG, we set $\alpha = 0.75$ and $c = \frac{1}{L_f}$. For a-IRG, we set $\eta_0 = 10^{-3}$ and $\gamma_0 = 10^{-3}$. For CG-BiO, we set $\gamma_k = 0.5/(k+1)$. For Bisec-BiO, we set the maximum number of iterations of the internal APG process to be $T' = 10000$. For FC-BiO$^{\text{sm}}$, FC-BiO$^{\text{Lip}}$ and Bisec-BiO, solving $\hat{g}^*$ takes 15000 iterations; for CG-BiO, solving $\hat{g}^*$ takes 10000 iterations. The results of such pretreatments are also plotted in Figure 1.

### E.2 Problem (12)

This problem is a $(L_f, L_g)$-smooth problem with

$$L_f = \frac{1}{4m}\lambda_{\max}((A^{val})^T A^{val}), \quad L_g = \frac{1}{4m}\lambda_{\max}((A^{tr})^T A^{tr}).$$

**Experiment setting**  In this experiment, a sample of 5000 instances is taken from the "rcv1.binary" dataset [8, 16] as the training set $(A^{tr}, \mathbf{b}^{tr})$; another 5000 instances is sampled as the validation set $(A^{val}, \mathbf{b}^{val})$. Each label $\mathbf{b}_i^{tr}$ (or $\mathbf{b}_i^{val}$) is either $+1$ or $-1$.

**Implementation details**  In this experiment, we set the initial point $\mathbf{x}_0 = 0$ for all methods. The implementation of our Algorithm 1 is similar to that in the first experiment. We set the maximum number of iterations of the subroutine to be $T' = 500$ and $T' = 1000$ for FC-BiO$^{\text{sm}}$ and FC-BiO$^{\text{Lip}}$ respectively. For FC-BiO$^{\text{Lip}}$, we set $\eta = 2$. For AGM-BiO, we set $\gamma = 1/(\frac{2L_g}{L_f}T^{\frac{2}{3}} + 2)$ as in [6, Theorem 4.4]. For PB-APG, we set $\gamma = 10^4$. For Bi-SG, we set $\alpha = 0.75$ and $c = \frac{1}{L_f}$. For a-IRG, we set $\eta_0 = 10^3$ and $\gamma_0 = 0.1$. For CG-BiO, we set $\gamma_k = \frac{2}{k+2}$. For FC-BiO$^{\text{sm}}$, solving $\hat{g}^*$ takes 1000 iterations; for FC-BiO$^{\text{Lip}}$ and CG-BiO, solving $\hat{g}^*$ takes 1500 iterations. As in the first experiment, the results of such pretreatments are also plotted in Figure 2.

