# OpenReview forum: "Functionally Constrained Algorithm Solves Convex Simple Bilevel Problem"
_NeurIPS.cc/2024/Conference — NeurIPS 2024 poster_

### Official Review · Reviewer_EaJT · 2024-07-09

**Soundness:** 3
**Presentation:** 3
**Contribution:** 2
**Rating:** 5
**Confidence:** 4

**Summary:**

The paper first shows the difficulties of obtaining the absolute optimal solutions to simple convex bilevel problems. The authors also present the lower bound of the first-order methods for solving simple convex bilevel problems. Moreover, the authors proposed a novel framework based on functionally constrained reformulation. Based on the framework, they developed near-optimal algorithms for solving simple convex bilevel problems under smooth and non-smooth settings.

**Strengths:**

1. The paper shows the difficulties of obtaining the absolute optimal solutions to simple convex bilevel problems under common assumptions.
2. The authors present the lower bounds of the first-order methods for solving simple convex bilevel problems, which are $\Omega(1/\sqrt{\epsilon})$ and $\Omega(1/\epsilon^{2})$ for smooth and non-smooth settings, respectively.
3. The authors also present the algorithms for solving these types of problems, which match the lower bounds ignoring the logarithmic factors.

**Weaknesses:**

1. Many lemmas in this paper are directly quoted from other works, but some, such as Lemma 5.4 and 5.5, are not identical to the original ones. It would be better to provide proof for these lemmas.
2. In this paper, the authors only consider the unconstrained case, while many other works about simple bilevel optimization consider the constrained case.
3. In this numerical experiment, the authors should count the time of finding initial $u$ and $l$ and show it on the plots. Otherwise, it is not a fair comparison, as the complexity of finding the initial $u$ and $l$ has the same complexity as the main stage.
4. The desired accuracy ($10^{-2}$ in Figures 1 and 2) for the experiments is not small enough. It would be better to set them to be smaller than $10^{-6}$ and see the performances of the algorithms for the late stage.
5. The proposed algorithm requires a predetermined $T$, whereas similar studies do not have this requirement.
6. The novelty of this work is limited, as the main idea is borrowed from [1] with some extensions, and many of the technical proofs are identical to [2] and [3].

References:

[1]. Jiulin Wang, Xu Shi, and Rujun Jiang. Near-optimal convex simple bilevel optimization with a bisection method.

[2]. Yurii Nesterov. Lectures on convex optimization

[3]. Sébastien Bubeck et al. Convex optimization: Algorithms and complexity.

**Questions:**

See weaknesses.

**Limitations:**

No negative societal impact.

---

> ### Author Rebuttal · Authors · 2024-08-07
>
> We express our gratitude to the reviewer for dedicating their valuable time and effort to evaluate our manuscript. Below are our responses to the reviewer’s concerns:
> ### W1
> Thanks for the suggestion! We will provide self-contained proofs for these lemmas in the revised version.
> ### W2
> Actually, our method can be extended to the constrained case where the feasible set is convex and compact (as in many previous works, e.g. [1][2]). In our manuscript, we focus on the unconstrained case so as to provide a clearer discussion.
>
> Specifically, if the feasible set of the problem is a convex and compact set $\mathcal X$, one only needs to replace the Euclidean ball $\mathcal B(\mathbf x_0,D)$ in the original problem with $\mathcal X$. For Lipschitz problems, computing a projection onto $\mathcal X$ is needed in each gradient step, which is standard. For smooth problems, one needs to compute the projection onto the intersection of $\mathcal X$ and a hyperplane (see Proposition 5.2), which is also needed by previous work [1].
> ### W3
> As presented in Appendix E, the initialization time is already taken into account in the numerical experiments and is shown in the plots. In the attached pdf in the general response, the initialization stage is plotted with dashed lines in Figure 1 and Figure 2.
> ### W4
> In the first experiment (Figure 1), we actually set the desired accuracy $\epsilon_f=\epsilon_g = 10^{-6}$. As we noted in Line 628-629, a $(\epsilon_f,\epsilon_g)$-weak optimal solution is indeed solved by $\texttt{FCB-BiO}^{\texttt{sm}}$, since the approximate solution $\hat {\mathbf x}$ found by the algorithm satisfies $g(\hat {\mathbf x })\leq g^*+\epsilon_g, f(\hat {\mathbf x})\leq f^*\leq f^*+\epsilon_f$. However, to present the results more clearly, Figure 1 shows the **absolute optimality gap** $|f(\hat {\mathbf x})-f^*|$, instead of the weak optimality gap $f(\hat{\mathbf x})-f^*$. Since the aim is to find a **weak optimal solution**, the absolute optimality gap does not converge to a very small level.
>
> The vertical axis of Figure 2 does not indicate the accuracy. As we remarked in Line 634-636, to the best of our knowledge, no existing solver can solve the exact optimal value $f^*,g^*$ of the second experiment. Thus we plot the function values $f(x),g(x)$, instead of the suboptimality gaps $f(x)-f^*$, $g(x)-g^*$ in Fugure 2.
>
> To better demonstrate the performances of the algorithms for the late stage, we conducted another numerical experiment. The setup and result of the experiment are presented in the general response and the attached pdf. In this experiment, we set $\epsilon_g=10^{-9}$ and $\epsilon_f=10^{-6}$. The results again show the superior performance of our method compared to existing methods in both upper-level and lower-level in the late stage.
> ### W5
> FCB-BiO is a double-loop algorithm (bisection as the first level and SGM/generalized AGM as the second level). We need to predetermine a $T$ (which is equivalent to predetermining a desired accuracy $\epsilon$) to stop the subprocess (SGM/generalized AGM) when it runs for enough time and the approximate solution is accurate enough.  Many previous double-loop methods also need to tackle such issues, such as Catalyst[5].
>
> We would like to remark that some previous works also need to predetermine the number of iterations $T$ to set the hyperparameters, e.g. learning rate, such as Theorem 4.4 in [1] and Proposition 1 in [3].
> ### W6
> We appreciate the reviewer's feedback and would like to clarify the distinctions and contributions of our work in comparison to previous works.
> Firstly, while the bisection process in our proposed algorithm $\texttt{FCB-BiO}$ might appear similar to the approach in Wang et. al. (2024) [4], the underlying motivations and objectives are fundamentally different. In Wang et. al. (2024) [4], the bisection is employed to transform the original problem $\min f(x) \text{ s.t.}\  g(x)\leq g^*$ to a series of  problems $\min \ g(x) \text{ s.t.}\  f(x)\leq c$ for different $c$. In contrast, in $\texttt{FCB-BiO}$, the original problem is reformulated to finding the smallest root of a univariate monotone auxiliary function $\psi^*(t)$. For such one-dimensional root-finding problems, the bisection method is the most standard and straightforward approach.
>
> Additionally, our reformulation improves the results in Wang et. al. (2024) [4].  Specifically, [4] requires an oracle that efficiently projects $x$ onto the sublevel set of $f$, i.e., {$x|f(x)\leq t$}. This assumption only holds for a limited class of functions and would otherwise incur additional gradient evaluations. On the other hand, $\texttt{FCB-BiO}$ does not need this assumption, making it applicable to a broader range of problems.
>
> In summary, we view the main contribution of our work as **introducing the functionally constrained reformulation** to simple bilevel problems and proposing a near-optimal method which is simple and effective. We also prove the intractability of finding absolute optimal solutions for zero-respecting first-order algorithms, and establish lower bounds for finding weak optimal solutions, ensuring the comprehensiveness of our study. The bisection search is only a subprocedure in our algorithm.
>
> [1] Cao, Jincheng, et al. An Accelerated Gradient Method for Simple Bilevel Optimization with Convex Lower-level Problem.
>
> [2] Jiang, Ruichen, et al. A conditional gradient-based method for simple bilevel optimization with convex lower-level problem.
>
> [3] Samadi S, et al. Achieving optimal complexity guarantees for a class of bilevel convex optimization problems.
>
> [4] Jiulin Wang, et al. Near-optimal convex simple bilevel optimization with a bisection method.
>
> [5] Lin H, et al. A universal catalyst for first-order optimization.
>
> We thank the reviewer once again for the effort and time invested in the review. We would love to provide further clarifications if the reviewer has any additional questions.

---

> ### Author Response · Authors · 2024-08-12
> **Could you kindly let us know if your concerns have been addressed?**
>
> Dear Reviewer EaJT:
>
> We sincerely appreciate the time and effort you have dedicated to reviewing our manuscript.  As the discussion period draws to a close, we kindly ask if you could let us know whether your concerns have been satisfactorily addressed. If you have any additional questions, we will be more than willing to provide further explanations.
>
> We would highly value any additional feedback you may provide.

---

> > ### Comment · Reviewer_EaJT · 2024-08-13
> >
> > Thank you for the rebuttal. I still believe the novelty of this paper is not significant. Both this paper and [Wang et al. (2024)] use an oracle of solving single-level problems and apply a bisection approach to tackle an associated 1-D problem. The only distinction is that this paper uses bisection to solve a different 1-D problem. Thus, I will maintain my score.

---

> > > ### Author Response · Authors · 2024-08-14
> > >
> > > Dear Reviewer EaJT,
> > >
> > > Thank you for your response. We greatly appreciate the time and effort you put into providing constructive feedback. We think your main concern is still the comparison to [1].
> > >
> > > We want to highlight that the sublevel set oracle (i.e. projecting onto the sublevel set of $f(\mathbf x)$: $\text{prox}_{f,c}(\mathbf x)=\arg \min _{f(\mathbf y)\leq c}\Vert\mathbf y-\mathbf x\Vert^2$) used in [1] is a **strong assumption**. For example, when $f(x)$ is the mean square loss or logistic loss as in our Experiment 2, projecting onto the sublevel set cannot be computed efficiently.
> > >
> > > In another example, we consider the single-level optimization of $f$ using such an oracle. In this case, one can simply solve this optimization problem using a binary search to identify $f^*$, which only requires $\mathcal{O}(\log(1/\epsilon))$ complexity to achieve $\vert f(x) - f^* \vert \le \epsilon$ (see also Remark 3 in \[1\]), even breaking the standard lower complexity bound $\Omega(1/\sqrt{\epsilon})$.
> > >
> > > In contrast, our method only uses the **standard gradient oracle** to achieve the $\tilde {\mathcal{O}}(1/ \sqrt{\epsilon})$ rate. We think this is a significant improvement over \[1\], and we hope that the reviewer can acknowledge our contributions.
> > >
> > > \[1\] Jiulin Wang, Xu Shi, and Rujun Jiang. Near-optimal convex simple bilevel optimization with a bisection method. International Conference on Artificial Intelligence and Statistics, 2024.

---

### Official Review · Reviewer_iPzZ · 2024-07-10

**Soundness:** 3
**Presentation:** 3
**Contribution:** 2
**Rating:** 5
**Confidence:** 4

**Summary:**

This work studies simple bilevel problems with convex upper and lower level objectives. The paper studies the problem for both smooth and Lipschitz functions. The contribution is two folded: 1- first, it shows that no zero-respecting algorithm can achieve $(\epsilon_f,\epsilon_g)$ absolute optimal solutions for limited number of iterations. 2- Second, it proposes the iteration complexity lower bound for first-order zero respecting algorithms and proposes a method that achieves this lower bound up to a logarithmic factor. The proposed method is achieved through relaxing Problem 1 to Problem 4 and solving this problem through a bisection procedure and Lemma 2.3.4 from [20]. Numerical experiments were conducted to study and compare the proposed method.

**Strengths:**

The paper has good literature review and good structure. I enjoyed reading it.

**Weaknesses:**

**Writing:**

1- paragraph starting from line 24: the first and second sentences are opposing each other. Meaning that the second sentence should bring something in contrast to what expressed in the first sentence. This is expected due to the use of "Yet". However, the comparison is not correct since the first sentence assumes an explicitly given constraint set while the second sentence is mentioning Problem 1 which does not have an explicit constraint set (which should be the case for bilevel programs).

2- In (5) it feels like a constraint over $x$ is missed in the minimization. Please clarify, thanks!

3- Line 205, the sentence starts with "And", but this sentence is connected to the previous sentence. Please fix!

4- Line 309, "It is open whether ..." sounds a bit informal, please revise, thanks!

5- Statements of Theorems 4.1, 4.2: "there exists an (1,1)..." should be "there exists a (1,1)..." to help readability.


**Technical:**

1- Discussion below Theorem 4.2 on its proof states that the failure of the algorithms happen when $n\geq 2T$ where $n$ is the problem dimension and $T$ is the number of calls to the first order oracle. This, however, was not mentioned anywhere else (at least I did not see it). This means that if you go beyond a number of calls to the oracle, then you can actually reach the approximate absolute optimal solutions and your first claim/contribution is falsified.

I'd like to add that the paper is studying the zero respecting property, which enforces coordinate-wise updates at each iteration. Convergence rate of $\mathcal O(n/T)$ is typical for coordinate methods when applied to convex problems. Thus, the dependency on $n$ (problem dimension) is expected naturally.

2- As a part of the algorithm, the initialization should be, though briefly, explained in the main body of the paper. I kindly suggest considering it.

3- The Bisec-Bio method showed a close behaviour in the first experiment (Figure 1). Due to the similar performance (not the same) between this method and the proposed method it is expected to see a more detailed comparison between these two in the second experiment. Unfortunately, this was skipped. More careful design of the second experiment might be helpful to increase the impact of the paper.

**Questions:**

Could you please explain your reasoning and ideas regarding the points raised in the "Weaknesses" section? Thanks!

**Limitations:**

yes

---

> ### Author Rebuttal · Authors · 2024-08-07
>
> We express our gratitude to the reviewer for dedicating their valuable time and effort to evaluate our manuscript. However, there are some misunderstandings in the review, and we respond to your concerns one by one.
>
> > Writing
>
> Thanks for the valuable suggestions! We will incorporate your advice in the revised version.
>
> > Technical Weakness 1: Theorem 4.1/4.2
>
> We respectfully disagree with your opinion. Throughout the paper, we adopt the standard oracle complexity framework for first-order methods established by Nesterov and  Nemirovski [1], [4]. A crucial aspect of this framework is that the function class is defined in a **dimension-free** manner (i.e. the dimension of the function can be arbitrarily large).  More specifically, in this framework, all **upper bounds** developed for gradient methods are dimension-free (which is a nature of gradient methods such as GD, SGD, and AGM) , and the **hardness results**, including lower complexity bounds (e.g. Theorem 5.1/5.2) and intractability results (e.g. Theorem 4.1/4.2) should also be dimension-free. Below we list some of such hardness results established in previous works. [1-3]
>
> Therefore, when proving hardness results, it is reasonable to state that for any fixed $T>0$, there exists a "hard problem", whose dimension can be large and depend on $T$, that is unsolvable in $T$ steps. In other words, we showed that no zero-respecting first-order methods can solve absolute optimal solutions to simple bilevel problems dimension-freely.
>
> We want to clarify that our hardness result applies to zero-respecting algorithms, which update all $n$ coordinates simultaneously. This is different from coordinate methods that only update one coordinate at each iteration.
>
> We also want to highlight that,  similar to our proof, most existing hardness results for first-order methods involve constructing "hard problems" whose dimension is larger than $T$. For example:
>
> * Standard lower bound for convex minimization (Theorem 2.1.7 [1].
> * Standard lower bound for nonconvex minimization (Theorem 1 [2]).
> * Recent intractability result for nonconvex-convex bilevel optimization (Theorem 3.2 [3]). They show that finding small hyper-gradients for nonconvex-convex bilevel optimization is hard for a fixed budget of $T$ iteration.
>
> [1] Nesterov, Yurii. Lectures on convex optimization. Vol. 137. Berlin: Springer, 2018.
>
> [2] Carmon, Y., Duchi, J. C., Hinder, O., & Sidford, A. (2020). Lower bounds for finding stationary points I. Mathematical Programming, 184(1), 71-120.
>
> [3]  Chen, L., Xu, J., & Zhang, J. On Finding Small Hyper-Gradients in Bilevel Optimization: Hardness Results and Improved Analysis. In COLT, 2024.
>
> [4] Nemirovskij A S, Yudin D B. Problem complexity and method efficiency in optimization. 1983.
>
>
> > Technical Weakness 2: As a part of the algorithm, the initialization should be, though briefly, explained in the main body of the paper.
>
> Thank you for the suggestion! We will add a concise explanation in the main body of the paper in the revised version.
>
> > Technical Weakness 3: Comparison with Bisec-BiO method in the experiment.
>
> The Bisec-BiO method has the same convergence rate as our proposed method FCB-BiO, thus shows a close behavior in the first experiment. However, we highlight that the advantage of FCB-BiO over Bisec-BiO is that its **applicability in a broader range of problems** (instead of a faster convergence on problems that both can solve). That's because Bisec-BiO requires a strong assumption that projection onto the sublevel set of the upper-level objective {$x|f(x)\leq a$} is easy to compute, while FCB-BiO **does not** rely on such an assumption. In the second experiment, the projection onto the sublevel set of the upper-level objective cannot be solved efficiently, rendering Bisec-BiO inapplicable.
>
> In summary, the first experiment was designed to illustrate a scenario where both methods are applicable and show comparable performance as expected. The second experiment, on the other hand, highlights a scenario where Bisec-BiO fails to apply,  yet our method is still applicable and shows near-optimal performance.
>
> We thank the reviewer once again for the effort and time invested in the review. We would love to provide further clarifications if the reviewer has any additional questions.

---

> ### Author Response · Authors · 2024-08-12
> **Could you kindly let us know if your concerns have been addressed?**
>
> Dear Reviewer iPzZ:
>
> We sincerely appreciate the time and effort you have dedicated to reviewing our manuscript.  As the discussion period draws to a close, we kindly ask if you could let us know whether your concerns, particularly regarding Theorem 4.1/4.2, have been satisfactorily addressed. If you have any additional questions, we will be more than willing to provide further explanations.
>
> We would highly value any additional feedback you may provide.

---

> > ### Comment · Reviewer_iPzZ · 2024-08-12
> >
> > Thanks for your responses. Considering your reponses to my comment and the Reviewer RLu3, I think the concerns are addressed. The score was adjusted accordingly.

---

### Official Review · Reviewer_KpB7 · 2024-07-11

**Soundness:** 3
**Presentation:** 3
**Contribution:** 3
**Rating:** 6
**Confidence:** 3

**Summary:**

This paper provides a theoretical proof that first-order zero-respecting algorithms are incapable of approximating the optimal solution for a simple bilevel optimization problem where both the upper-level and lower-level functions are convex. Then they propose a functional constrained reformulation to solve the simple bilevel problem. The method achieves the near-optimal convergence rate and shows good experimental results.

**Strengths:**

- This paper provides the theoretical proof to show that the zero-respecting first-order methods cannot find absolute optimal solutions.

- They proposed a novel algorithm FCB-BiO to find $(\epsilon_f, \epsilon_g)$-weak optimal solutions for non-smooth and smooth cases, and this method achieves the near-optimal rates.

- Compared to [1], this paper gets rid of the weak sharp minima condition.

[1] Cao, Jincheng, et al. "An Accelerated Gradient Method for Simple Bilevel Optimization with Convex Lower-level Problem." arXiv preprint arXiv:2402.08097 (2024).

**Weaknesses:**

- Can other baselines like [1], [2] find $(\epsilon_f, \epsilon_g)$-weak optimal solutions? Actually, they can find absolute solutions with additional conditions, but hard to say they cannot work well for weak optimal solutions.

- The experiments are little bit weak, and they simply specify $f(.)$ and $g(.)$ as quadratic functions. Can it be applied to neural networks? For example, $x$ is the model parameters of a neural network.

- It would be better if authors can show the results on meta-learning or some other typical bilevel optimization problems.

- For Figure 1 and 2, do they take the warm-up time and initialization time into consideration?

[1] Cao, Jincheng, et al. "An Accelerated Gradient Method for Simple Bilevel Optimization with Convex Lower-level Problem." arXiv preprint arXiv:2402.08097 (2024).

[2] Jiang, Ruichen, et al. "A conditional gradient-based method for simple bilevel optimization with convex lower-level problem." International Conference on Artificial Intelligence and Statistics. PMLR, 2023.

**Questions:**

Please refer to the weaknesses.

**Limitations:**

Yes.

---

> ### Author Rebuttal · Authors · 2024-08-07
>
> We express our gratitude to the reviewer for dedicating their valuable time and effort to evaluate our manuscript. Below are our responses to the reviewer’s concerns.
>
> > W1: Can other baselines like [1], [2] find $(\epsilon_f,\epsilon_g)$ weak optimal solutions?
>
> Actually, while some of the previous works have developed algorithms to find absolute solutions under additional assumptions, most previous methods, including [1], [2], aim to find weak optimal solutions. So yes, they can find weak optimal solutions. However, as we discussed in the "Related work" section, previous methods either do not achieve optimal convergence rate (e.g. [2], which only achieves a suboptimal $O(1/\epsilon)$ rate for smooth problems), or require additional assumptions such as weak sharp minima condition (e.g. [1]). In contrast, our proposed method achieves near-optimal rates without needing such assumptions.
>
> [1] Cao, Jincheng, et al. "An Accelerated Gradient Method for Simple Bilevel Optimization with Convex Lower-level Problem." arXiv preprint arXiv:2402.08097 (2024).
>
> [2] Jiang, Ruichen, et al. "A conditional gradient-based method for simple bilevel optimization with convex lower-level problem." International Conference on Artificial Intelligence and Statistics. PMLR, 2023.
>
> > W2: The experiments are little bit weak, and they simply specify $f$ and $g$  as quadratic functions. Can it be applied to neural networks?
>
> We would like to clarify that in our second experiment, the objective functions $f,g$ are the loss functions of logistic regression, rather than quadratic functions. We acknowledge that our method is designed for convex simple bilevel optimization problems as in many previous works, thus is not directly applicable to neural networks which are nonconvex. However, we would like to highlight that there is **no existing theory** for simple bilevel problems with nonconvex lower-level objectives.  But we are willing to study it in the future works.
>
> > W3: It would be better if authors can show the results on meta-learning or some other typical bilevel optimization problems.
>
>
> Meta-learning takes the following form:
> $$
> \begin{align*}
> \min_{x,y}\ & f(y)\\\\
> {\rm s.t.} \quad y\in & \arg\min_z g(z)+\frac \lambda 2 \|x-z\|^2
> \end{align*}
> $$
> which is a more general bilevel setting where the lower-level problem is parameterized by the upper-level variable $x$. See Equation 4 in [3]. This is different from our setup of simple bilevel optimization (Equation 1 in the manuscript).
>
> [3] Rajeswaran, A., Finn, C., Kakade, S. M., & Levine, S. Meta-learning with implicit gradients. In NeurIPS, 2019.
>
> > W4: For Figure 1 and 2, do they take the warm-up time and initialization time into consideration?
>
> Yes, as discussed in Appendix E, the initialization time is also taken into account and is plotted in Figure 1 and Figure 2 of the manuscript. In the attached pdf in the general response, the initialization stage is plotted with dashed lines in Figure 1 and Figure 2.
>
> We thank the reviewer once again for the effort and time invested in the review. We would love to provide further clarifications if the reviewer has any additional questions.

---

> ### Author Response · Authors · 2024-08-12
> **Could you kindly let us know if your concerns have been addressed?**
>
> Dear Reviewer KpB7:
>
> We sincerely appreciate the time and effort you have dedicated to reviewing our manuscript.  As the discussion period draws to a close, we kindly ask if you could let us know whether your concerns have been satisfactorily addressed. If you have any additional questions, we will be more than willing to provide further explanations.
>
> We would highly value any additional feedback you may provide.

---

> > ### Comment · Reviewer_KpB7 · 2024-08-12
> >
> > Thanks for your response. I would like to keep my rating.

---

### Official Review · Reviewer_213i · 2024-07-11

**Soundness:** 4
**Presentation:** 3
**Contribution:** 3
**Rating:** 6
**Confidence:** 4

**Summary:**

This work proposes a novel and near-optimal method to solve convex simple bilevel problems by finding weak optimal solutions. The author also provides theoretical and numerical guarantees of the convergence of this algorithm.

**Strengths:**

1. This paper is easy to follow, with techniques that are rigorously proven and thoroughly explained.

2. This work is proven without the Hölderian error bound condition, and the lower bounds clearly demonstrate near-optimality in both Lipschitz and smooth settings.

**Weaknesses:**

1. Since this work focuses only on weak optimal solutions, the title *Near-Optimal Methods for Convex Simple Bilevel Problems* seems somewhat boastful.

2. There are some minor typos like missing space before "When" in line 238 and 261.

**Questions:**

1. What will happen if one of $f$ and $g$ is Lipschitz continuous and the other one is smooth?

2. What it is "," in eq.(5) but ";" in eq.(6)?

3. Assumption 3.1.2 does not appear in previous works and seems very strong. Could the author further explain why this assumption is necessary and what the difficulties would be without it?

4. How does the author determine the initial interval $[l, u]$, and how does this interval affect the convergence?

I would like to change my grade based on the response from the author.

**Limitations:**

No limitation

---

> ### Author Rebuttal · Authors · 2024-08-07
>
> We express our gratitude to the reviewer for dedicating their valuable time and effort to evaluate our manuscript. Below are our responses to the reviewer’s concerns.
>
> > W1: Since this work focuses only on weak optimal solutions, the title Near-Optimal Methods for Convex Simple Bilevel Problems seems somewhat boastful.
>
>
> In this paper, we study two solution concepts, absolute optimal solutions and weak optimal solutions. However, when we say "near-optimal", we refer specifically to weak optimal solutions, as we have proven that absolute optimal solutions are intractable.
>
>
> We aim to have a concise title that accurately reflects the focus of our work, and we are open to suggestions from the reviewers to further refine and improve the title.
>
> > W2: minor typos.
>
> Thanks for pointing out the typos. We will carefully review and correct the typos in the revised version.
>
> > Q1: What will happen if one of $f$ and $g$ is Lipschitz continuous and the other one is smooth?
>
> This is an interesting and valuable question. A similar intractability result of finding absolute optimal solutions as presented in Section 4 still holds in this case. The proof would be similar to the one of Theorem 4.1/4.2, by letting the lower-level objective g be a Lipschitz/smooth zero-chain and letting the upper-level function f be a quadratic function that only depends on the last half of the components.
>
> A similar lower bound for finding weak optimal solutions as in Section 5.1 can also be established, as the argument of decoupling the optimization processes for $f$ and $g$ still applies. More specifically, the lower bound would be $\Omega \left(\max\left(\frac{L_f}{\epsilon_f}D, \frac{C_g^2}{\epsilon_g^2}D^2\right)\right)$ if $f$ is $L_f$-smooth and $g$ is $C_g$-Lipschitz, or $\Omega \left(\max\left(\frac{C_f^2}{\epsilon_f^2}D^2,\frac{L_g}{\epsilon_g}D\right)\right)$ if $f$ is $C_f$-Lipschitz and $g$ is $L_g$-smooth.
>
> For the upper bound, a smooth function is also Lipschitz-continuous in $\mathcal B (\mathbf x_0,D)$, thus applying $\texttt{FCB-BiO}^{\texttt{Lip}}$ directly gives the complexity of $\tilde O(\max\{\frac{1}{\epsilon_f^2},\frac 1 {\epsilon_g^2}\})$. If $\epsilon_f\asymp \epsilon_g$, then the dependency on $\epsilon$ is near-optimal. However, achieving near-optimal dependency on other parameters remains an open question and would be an interesting direction for future research.
>
> > Q2: What it is "," in eq.(5) but ";" in eq.(6)?
>
> This is a typo, and we apologize for any confusion it caused. The notation in (5) and (6) should be consistent and both signify the maximum of $f(\mathbf x)-t$ and $\tilde g(\mathbf x)$
>
> > Q3: Assumption 3.1.2 does not appear in previous works and seems very strong. Could the author further explain why this assumption is necessary and what the difficulties would be without it?
>
> The assumption that $\|\mathbf x-\mathbf x_0\|\leq D$ is a technical assumption due to the bisection procedure. This procedure involves solving a series of subproblems $\text{minimize} \max(f(\mathbf x)-t,\tilde g(\mathbf x))$ with resect to $\mathbf x$ (Equation 6), the total complexity would depend on $\max_t \|\mathbf x^*_{(t)}-\bar {\mathbf x}\|$, where $\bar {\mathbf x}$ is the starting point of the subprocess, and $\mathbf x_{(t)}^*$ is the optimal solution of the subproblem. The term $\|\mathbf x^*_{(t)}-\bar {\mathbf x}\|$ cannot be bounded by the initial distance $\|\mathbf x^*-\mathbf x_0\|$ because $\mathbf x^*_{(t)}$ might be far away from $\mathbf x^*$. To address this, we restrict the domain to $\mathcal B(\mathbf x_0,D)$, ensuring that the distance term $\|\mathbf x^*_{(t)}-\bar {\mathbf x}\|$ is always bounded by $D$ in each bisection step.
>
> > Q4: How to determine the initial interval $[\ell, u]$, and how does this interval affect the convergence?
>
> As mentioned in Line 201, we discuss how to determine the initial interval $[\ell,u]$ in Appendix B.1. Specifically, we apply single-level optimization methods SGM/AGM to $f$ and $g$ respectively and solve approximate solutions $\hat {\mathbf x}_f,\hat {\mathbf x}_g$ such that
> $$
> p^*\leq f(\hat {\mathbf x} _f)\leq p^*+\frac \epsilon 2,g^*\leq g(\hat {\mathbf x} _g)\leq g^*+\frac \epsilon 2,
> $$
> where $p^*,g^*$ are global minima of $f$ and $g$ respectively. Then setting $\ell =  f(\hat {\mathbf x} _f)-\frac \epsilon 2\leq p^*\leq \hat f^*$ would be a valid lower bound of $\hat f ^*$, and setting  $u=f(\hat {\mathbf x}_g)$ would be a valid upper bound.
>
> As presented in Theorem 5.3 and Theorem 5.4, the initial distance $u-\ell$ affects the convergence rate up to a logarithm term. If $f$ is bounded, then $u-\ell$ is also bounded.
>
> We thank the reviewer once again for the valuable and helpful suggestions. We would love to provide further clarifications if the reviewer has any additional questions.

---

> > ### Comment · Reviewer_213i · 2024-08-12
> >
> > Thank you for the authors' response and efforts. I'd prefer to maintain my current score. Additionally, it would be great to see a relaxed assumption in place of Assumption 3.1.2.

---

> ### Author Response · Authors · 2024-08-12
> **Could you kindly let us know if your concerns have been addressed?**
>
> Dear Reviewer 213i:
>
> We sincerely appreciate the time and effort you have dedicated to reviewing our manuscript.  As the discussion period draws to a close, we kindly ask if you could let us know whether your concerns have been satisfactorily addressed. If you have any additional questions, we will be more than willing to provide further explanations.
>
> We would highly value any additional feedback you may provide.

---

### Official Review · Reviewer_RLu3 · 2024-07-27

**Soundness:** 2
**Presentation:** 2
**Contribution:** 2
**Rating:** 6
**Confidence:** 4

**Summary:**

The paper proposes a method to solve the bilevel problems where both upper and lower level objective functions are convex. The new algorithms, $FCB-BiO^{sm}$ combine bisection with sub-gradient or gradient methods to solve a reformulated problems to find a weak optimal solution. Convergence analyses show that the proposed method achieves the optimal rate up to a log factor. Numerical experiments illustrate the performance of $FCB-BiO^{sm}$ compared to existing works.

**Strengths:**

- Different from other related work, the paper focuses on the reformulated minimax problem of the original bilevel optimization.
- To solve the reformulated minimax problem, the paper uses bisection followed by gradient or sub-gradient methods.
- The assumption in the paper is more relaxed than existing works that use Hölderian error bound condition.

**Weaknesses:**

- Theorem 4.1 and Theorem 4.2 are not well-motivated and unclear to me. The theorems state that for any algorithm that runs for a fixed budget of T iteration, there exists a problem instance such that we cannot achieve the absolute optimal solution. By showing this, the authors claim that finding absolute optimal solutions for the class of bilevel problem (1) is hard which I disagree. The key point is that, since we only run for T iterations, what if we did not run enough iterations so that the absolute optimal solution is achieved. As shown in the proof of Theorem 4.1 and Theorem 4.2, the authors come up with a problem of dimention $2T$ where the algorithms only run up to T iteration and in each iteration, only 1 component becomes non-zero. What if for the same problem, we allocate more budget of $\hat{T}$ iteration where $\hat{T} >> T$?
- Also, showing that there exists one instance such that when running an algorithms for T iteration cannot lead to absolute optimal solution within T iteration does not mean that the problem cannot be solved for any T iteration for $T\to\infty$.
- Due to the above concerns, the paper does not containt much novelty as the bisection technique has already been proposed in Wang et. al. (2024) [30], and the convergence result of the gradient/sub-gradient methods are known.

Minor comments:
- Typo at line 279: "minimmum" $\to$ "minimum"?

**Questions:**

1. To come up with an intial interval $[\ell ,u]$, we need to run SGM or AGM applied to $g(x)$. However, to do so, we require knowledge of the Lipschitz constant which might not always be available. How can we come up with the interval in such cases?

**Limitations:**

The paper does discuss its limitation.

---

> ### Author Rebuttal · Authors · 2024-08-07
>
> We express our gratitude to the reviewer for dedicating their valuable time and effort to evaluate our manuscript. However, there are some misunderstandings in the review that we wish to clarify.
>
> > W1 & W2: In the proof of Theorem 4.1/4.2, what if for the same problem, we allocate more budget of iterations, so that the absolute optimal solution is achieved?
>
>
> We respectfully disagree with your opinion. Throughout the paper, we adopt the standard oracle complexity framework for first-order methods established by Nesterov and  Nemirovski [1], [4]. A crucial aspect of this framework is that the function class is defined in a **dimension-free** manner (i.e. the dimension of the function can be arbitrarily large).  More specifically, in this framework, all **upper bounds** developed for gradient methods are dimension-free (which is a nature of gradient methods such as GD, SGD, and AGM) , and the **hardness results**, including lower complexity bounds (e.g. Theorem 5.1/5.2) and intractability results (e.g. Theorem 4.1/4.2) should also be dimension-free. Below we list some of such hardness results established in previous works. [1-3]
>
> Therefore, when proving hardness results, it is reasonable to state that for any fixed $T>0$, there exists a "hard problem", whose dimension can be large and depend on $T$, that is unsolvable in $T$ steps. In other words, we showed that no zero-respecting first-order methods can solve absolute optimal solutions to simple bilevel problems dimension-freely.
>
> We also want to highlight that,  similar to our proof, most existing hardness results for first-order methods involve constructing "hard problems" whose dimension is larger than $T$. For example:
>
> * Standard lower bound for convex minimization (Theorem 2.1.7 [1]).
> * Standard lower bound for nonconvex minimization (Theorem 1 [2]).
> * Recent intractability result for nonconvex-convex bilevel optimization (Theorem 3.2 [3]). They show that finding small hyper-gradients for nonconvex-convex bilevel optimization is hard for a fixed budget of $T$ iterations.
>
> [1] Nesterov, Yurii. Lectures on convex optimization. Vol. 137. Berlin: Springer, 2018.
>
> [2] Carmon, Y., Duchi, J. C., Hinder, O., & Sidford, A. (2020). Lower bounds for finding stationary points I. Mathematical Programming, 184(1), 71-120.
>
> [3]  Chen, L., Xu, J., & Zhang, J. On Finding Small Hyper-Gradients in Bilevel Optimization: Hardness Results and Improved Analysis. In COLT, 2024.
>
> [4] Nemirovskij A. S., Yudin D. B. Problem complexity and method efficiency in optimization. 1983.
>
>
> > W3: Novelty; Similarity with Wang et. al. (2024) [1]
>
> We appreciate the reviewer's feedback. We have addressed the rationality of Theorem 4.1/4.2 in our previous discussion. Below we would like to clarify the distinctions and contributions of our work in comparison to previous works.
>
> Firstly, while the bisection process in our proposed algorithm $\texttt{FCB-BiO}$ might appear similar to the approach in Wang et. al. (2024) [1], the underlying motivations and objectives are fundamentally different. In Wang et. al. (2024) [1], the bisection is employed to transform the original problem $\min f(x) \ \text{s.t.}\  g(x)\leq g^*$ to a series of  problems $\min \ g(x) \ \text{s.t.}\  f(x)\leq c$ for different $c$. In contrast, in $\texttt{FCB-BiO}$, the original problem is reformulated to finding the smallest root of a univariate monotone auxiliary function $\psi^*(t)$. For such one-dimensional root-finding problems, the bisection method is the most standard and straightforward approach.
>
> Additionally, our reformulation improves the results in Wang et. al. (2024) [1].  Specifically, [1] requires an oracle that efficiently projects $x$ onto the sublevel set of $f$, i.e., {$x|f(x)\leq t$}. This assumption only holds for a limited class of functions and would otherwise incur additional gradient evaluations. On the other hand, $\texttt{FCB-BiO}$ does not need this assumption, making it applicable to a broader range of problems.
>
>
> In summary, we view the main contribution of our work as **introducing the functionally constrained reformulation** to simple bilevel problems and proposing near-optimal method which is simple and effective. We also prove the intractability of finding absolute optimal solutions for zero-respecting first-order algorithms, and establish lower bounds for finding weak optimal solutions, ensuring the comprehensiveness of our study. The bisection search is only a subprocedure in our algorithm.
>
> [1]. Jiulin Wang, Xu Shi, and Rujun Jiang. Near-optimal convex simple bilevel optimization with a bisection method. International Conference on Artificial Intelligence and Statistics, 2024.
>
> > Minor Comments
>
> Thanks for pointing out the typo. We will fix it in the revised version.
>
>
> > Q1: To come up with an initial interval $[\ell,u]$，we need to run SGM or AGM applied to $g(x)$. This requires the knowledge of Lipschitz constant which might not always be available.
>
> There are two approaches we can take to tackle this problem:
>
> First, tune the Lipshitz constant as a hyperparameter, which is also equivalent to tuning the learning rates.
>
> Second, apply a parameter-free method on $g(x)$, such as [1,2].
>
> [1]  Guanghui Lan, Yuyuan Ouyang, and Zhe Zhang. "Optimal and parameter-free gradient minimization methods for smooth optimization." arXiv preprint arXiv:2310.12139 (2023).
> [2] Itai Kreisler, Maor Ivgi, Oliver Hinder, and Yair Carmon. "Accelerated Parameter-Free Stochastic Optimization." In COLT, 2024.
>
> We thank the reviewer once again for the effort and time invested in the review. We would love to provide further clarifications if the reviewer has any additional questions.

---

> ### Author Response · Authors · 2024-08-12
> **Could you kindly let us know if your concerns have been addressed?**
>
> Dear Reviewer RLu3:
>
> We sincerely appreciate the time and effort you have dedicated to reviewing our manuscript.  As the discussion period draws to a close, we kindly ask if you could let us know whether your concerns, particularly regarding Theorem 4.1/4.2, have been satisfactorily addressed. If you have any additional questions, we will be more than willing to provide further explanations.
>
> We would highly value any additional feedback you may provide.

---

> > ### Comment · Reviewer_RLu3 · 2024-08-12
> > **Response to authors**
> >
> > Dear authors,
> >
> > Thank you for providing responses to my comments. I have additional comments below:
> >
> > - I believe that the hardness results in [3] is the closest setting to this paper where they show that under zero-chain assumption, the solution stays at 0 at any iteration.
> > - Line 149-154 in the paper confused me at the beginnning mainly from the statement "Due to the zero-respecting property, only one component of $x_k$ is activated (i.e. becomes nonzero) per iteration." If this happens, then at a certain iteration large enough, we would be able to arrive at the optimal solution. However, in the proof the authors actually prove that the solution stays at 0 if the initial point is 0. I suggest clarifying this in the paper.
> >
> > As my concerns have been mostly addressed, I have adjusted the score accordingly.

---

> > > ### Author Response · Authors · 2024-08-14
> > >
> > > Dear Reviewer RLu3,
> > >
> > > Thank you for your thoughtful feedback and for adjusting the score. We appreciate your additional comments and will clarify the points you mentioned in the revised version.

---

### Author Rebuttal · Authors · 2024-08-07

We are deeply grateful for the efforts and valuable feedback of the reviewers and area chairs in reviewing our manuscript.

Combining the suggestions of the reviewers, we conduct an additional numerical experiment to better compare the performance of $\texttt{FCB-BiO}$ with other methods. Following previous works [1],[2], we consider the over-parameterized linear regression problem. We use the Wikipedia Math Essential dataset [24], which contains $1068$ instances with $730$ attributes.  We use $50\%$ of the dataset as the training set $(A^{tr},\mathbf b^{tr})$ and $50\%$ of the dataset as the validation dataset $(A^{val},\mathbf b^{val})$. The problem can be formulated as:

$$
\begin{aligned}
\min f(\mathbf x)&:=\frac 1{2m^{val}} \|A^{val}\mathbf x-\mathbf b^{val}\|^2,\\\\
\text{s.t.}\quad \mathbf x&\in \arg\min_{\mathbf z}g(\mathbf z):=\frac 1 {2m^{tr}}\|A^{tr}\mathbf z -\mathbf b^{tr}\|^2.
\end{aligned}
$$

The result is shown in Figure 3 in the uploaded pdf. The initialization time is plotted with dashed lines.

In this experiment, we set $\epsilon_g=10^{-9}$ and $\epsilon_f=10^{-6}$. The optimal solution of this problem can be solved by Lagrange multiplier method. The result again shows the superior performance of our method compared to existing methods in both upper-level and lower-level. We remark that in this experiment, an $(\epsilon_f,\epsilon_g)$-**absolute optimal solution** is solved by $\texttt{FCB-BiO}$. But this is not a contradiction to our hardness result in Section 4. For this problem, the lower-level objective satisfies the second-order Holderian Error Bound condition. As we discussed in Appendix D, $\texttt{FCB-BiO}$ can solve the absolute optimal solution with the best known rate when the lower-level objective satisfies this additional Holderian error bound condition.

In this experiment, we set the initial point $\mathbf x_0$ to be a random vector of unit length. For $\texttt{AGM-BiO}$, we set $\gamma = 1 /(\frac {2L_g}{L_f}T^{2/3}+2)$ as in Theorem 4.4 [1]. For $\texttt{PB-APG}$, we set $\gamma = 10^5$. For $\texttt{a-IRG}$, we set $\eta_0=1$ and $\gamma_0=10^{-3}$. For $\texttt{CG-BiO}$, we set $\gamma_k = \frac 2 {k+1}$.

[1] Cao, Jincheng, et al. "An Accelerated Gradient Method for Simple Bilevel Optimization with Convex Lower-level Problem." arXiv preprint arXiv:2402.08097 (2024).

[2] Jiang, Ruichen, et al. "A conditional gradient-based method for simple bilevel optimization with convex lower-level problem." International Conference on Artificial Intelligence and Statistics. PMLR, 2023.

---

### Decision · Program_Chairs · 2024-09-25

**Decision:**

Accept (poster)

**Comment:**

Simple bilinear optimization is the problem of minimizing a function f(x) subject to the constraint that x \in argmin g(x) where f and g are convex functions. This problem is equivalent to solving min f(x) s.t. g(x) <= min_z g(z) (i.e., it is a nonlinear constrained optimization problem with a single constraint).

This paper combines ideas from several different parts of the optimization literature to resolve the complexity of simple bilinear optimization. Experimental results of the proposed algorithm on finding the minimum norm solution of a linear system and overparameterized linear models indicate it is potentially practical. The reviewers appreciated that the paper was well written. There were some concerns raise by the reviewers EaJT about the limited novelty of the work given that it primarily specializes ideas from Nesterov's convex optimization lecture notes to simple bilinear optimization.

Comments:
-  Definitions of zero-respecting and zero chains are not correct. These terms are defined originally in [8] and are generalization of the ideas of Nesterov, please use them correctly or use other terms.
- It should be made crystal clear in the text that Theorem 5.1 and Theorem 5.2 follows immediately from existing lower bounds. Also the proof can be simplified: it suffices simply to observe which of L_f and L_g is larger and then set the corresponding function to the standard lower bound and the other function to zero.